

# A modeling methodology to study the tributary-junction alluvial fan connectivity during a debris flow event

Alex Garcés[1], Gerardo Zegers[2], Albert Cabré[1], Germán Aguilar[1], Aldo Tamburrino[1,3], and Santiago Montserrat[1]

[1]Advanced Mining Technology Center, Universidad de Chile
[2]Department of Geosciences, University of Calgary
[3]Department of Civil Engineering, Universidad de Chile

**Correspondence:** Santiago Montserrat (santiago.montserrat@amtc.cl)

**Abstract.** Traditionally, interactions between tributary alluvial fans and the main river have been studied on the field and in the laboratory, giving rise to different conceptual models explaining its role in the sediment cascade. On the other hand, numerical modeling of these complex interactions is still limited because the broad debris flow transport regimes are associated with different sediment transport models. Even though sophisticated models capable of simulating many transport mechanisms simultaneously exist, they are restricted to research purposes due to their high computational cost. In this article, we propose a workflow to model the response of an alluvial fan in the Huasco Valley, located in the Atacama Desert, during an extreme storm event. For the Crucecita Alta alluvial fan, five different deposits were identified and associated with different debris flow surges. Using a commercial software, our workflow concatenates these surges into one model. This study depicts the significance of the mechanical classification of debris flows to reproduce how an alluvial fan controls the tributary-river junction connectivity. Once our model is calibrated, we use our workflow to test if a channel is enough to mitigate the impacts of these flows and the effects on the tributary-river junction connectivity.

## 1 Introduction

In arid and semi arid regions, extreme storm events commonly trigger debris flows with high potential to modify the landscape (Mather and Hartley, 2005; Michaelides and Singer, 2014). The sediment stored in the catchments is transported towards tributary-junction alluvial fans and main rivers during these events. Tributary-junction alluvial fans play a crucial role in the transference of sediment from source areas to the main river (Fryirs, 2013; Aguilar et al., 2020). The efficiency of this sediment transference depends on the degree of connectivity within the fluvial system (Brunsden and Thornes, 1979), where tributary-junction alluvial fans can modulate the sediment transference towards the main river by buffering or bypassing the sediment discharge, creating different coupling conditions (Heckmann et al., 2018; Savi et al., 2020) that depend on the combination of the fan's geomorphological configuration and the flow properties. The degree of connectivity between tributaries and the main river determines the down system transmission of sediments and water and, consequently, its sensitivity to any environmental change (Mather et al., 2017; Cabré et al., 2020a). Understanding the fan dynamics and fan-river interactions is necessary to





comprehend the sediment cascade during debris flow events. However, these processes are still not properly modeled in hazard assessment projects (Savi et al., 2020).

The alluvial fan's geomorphic configuration may help foresee their coupling conditions. For example, mild slopes fans, or the absence of a feeder channel connecting the tributary with the main river, can promote sediment deposition, thus buffering the sediment discharge (Mather et al., 2017; Zegers et al., 2017). Hence, alluvial fans can act as a sediment decoupler feature in the sediment cascade, preventing lateral sediment discharges from reaching the main river (Fryirs et al., 2007). Contrary, alluvial fans with a trimmed fan toe, known as "toe-cutting" (Leeder and Mack, 2001), enhance alluvial fan trenching and

consequent generation of lobes at the fan toe. Telescopic-like deposit morphologies result from a local increase of the sediment transport in the alluvial fan, followed by an alluvial fan progradation (Colombo, 2005). In this case, the lateral sediment discharge becomes completely coupled with the river. However, when this progradation causes a river blockage, it acts as a barrier by decoupling the longitudinal sediment discharge along the river (Fryirs et al., 2007). The alluvial fan's geomorphic configuration and its connectivity can also be affected by anthropogenic activities and infrastructure. For example, in debris

flows prone areas, hydraulics works are used to retain sediments (check-dams) and/or to safely conduct the flows (channels). These works artificially change the coupling condition between the tributary fan and the river.

    The interaction between tributary alluvial fans with the main river has been studied on the field (Mather et al., 2017; Cabré et al., 2020a) and in the laboratory (Savi et al., 2020), giving rise to different conceptual models explaining its role in the sediment cascade (Mather et al., 2017; Cabré et al., 2020a; Savi et al., 2020). On the other hand, numerical simulation is

limited because available models cannot tackle all the possible flows types and sediment transport processes occurring in an alluvial fan and its interaction with a river. Some approaches have integrated many essential physical phenomena into one generalized debris flow model (Pudasaini, 2012; Mergili et al., 2018). However, the lack of sound field evidence has limited the validation of these models.

    Debris flows are influenced by forces arising from particle-particle, fluid-particle, and fluid-fluid interactions (Iverson, 1997).

When the interstitial fluid is a slurry composed of a high proportion of fine sediments, the density difference between the fluid and particles is small, allowing coarse sediment to flow at the same velocity as the slurry (Takahashi, 2014). These flowing mixtures are known as viscous debris flows and are well represented by a single-phase viscoplastic rheological model such as Bingham or Herschel-Bulkley rheologies (Takahashi, 2014; Naef et al., 2006; Montserrat et al., 2012). The turbulent stress dominates the flow behaviour for higher velocities, and Manning or Chézy type relations provide good results (Naef et al., 2006;

?). Conversely, when the interstitial fluid is mainly water, coarse sediment and water move at different relative velocities (von Boetticher et al., 2016). In this case, grain collisions, or dispersive stress, dominate the bulk flow energy dissipation (Takahashi, 2014; Naef et al., 2006). Similar to turbulent stress, collisional stresses are proportional to the square velocity (inertial stress); thus, pseudo-Manning or Chézy types approaches have been used to estimate bulk stress (Naef et al., 2006; Rickenmann et al., 2006). In these approaches, pseudo-Manning or Chézy coefficients could be functions of grain types and size, volume

concentration, among others (Naef et al., 2006; Rickenmann et al., 2006). For slow and/or highly concentrated flows, instead of collisions, particles can experience long-lasting contacts. In this case, the Coulomb plasticity model has shown to be a good approach for modeling interparticle frictional stress (Ancey, 2007; Montserrat et al., 2012). Coulomb stress operates similarly





to yield stress in Bingham or Herschel-Bulkley rheological models; thus, it can be surrogated. Coulomb or yield stress allows debris flows to stop in numerical models, reproducing deposits. Existing flow resistance relationships for debris-flow modeling

basically combine viscous, yield/coulomb, and turbulent/dispersive stress in a single equation accounting for bulk frictional losses (Naef et al., 2006).

One-phase models are preferred when representing bed material entrainment and deposition in numerical models due to computational costs. The incorporation of sediment in the flow produces changes in the rheological properties and an increase in the volume of the mixture (Iverson, 1997; Naef et al., 2006; Zegers et al., 2020). Entrainment and deposition continue to

be pivotal components for modeling and reproducing the topographic adjustments during debris flow events (Cao et al., 2004). However, there is still no consensus on the parameters that govern the sediment entrainment. For example, Takahashi et al. (1992) proposed a method that incorporates sediment into the flow until reaching an equilibrium concentration $C_\infty$. Cao et al. (2004) proposed entrainment and deposition relationships as a function of the shear stress, volume concentration, size of the particles, and the flow height and velocity, together with three coefficients that need to be calibrated. McDougall and Hungr

(2005) represent sediment entrainment with an exponential growth parameter $E$, which is independent of the flow velocity. Since none of these models reproduces the physical phenomena properly, it is recommended to use the expressions with fewer parameters that need to be calibrated (McDougall and Hungr, 2005).

In March 2015, an extreme rainstorm event occurred over a large area of the Chilean Atacama Desert (Bozkurt et al., 2016; Wilcox et al., 2016; Jordan et al., 2019; Cabré et al., 2020b), hereinafter called the 25M event. This study benefits from a

detailed sedimentological field characterization of the debris flow deposits on the *Crucecita Alta* fan for the 25M event (Cabré et al., 2020a). The *Crucecita Alta* catchment is a tributary of the *El Carmen* River in the *Huasco* Valley (∼29°S, 70°W). The conceptual connectivity model proposed by Cabré et al. (2020a) is based on the characteristic storm signature of the 25M event, which resulted in characteristic debris flow surges. We used this data to differentiate two main sediment transport modes and assigned them to different sediment transport models for each surge. We developed a workflow that includes the calibration

of the flow's rheological parameters and a python routine to concatenate different numerical models for the different debris flow surges. We used the geomorphic changes that occurred in *Crucecita Alta* to test the suitability of our workflow in the reproduction of the 25M event on this alluvial fan.

Inhabitants of the *Huasco* Valley tend to dwell in the alluvial fans because of their gentle slopes and because alluvial fans are safe places during river floods. However, low-frequency high-magnitude debris flows events directly impact these populated

areas. To reduce the impacts of debris flows, mitigation works in such environments are divided into sediment retention (pools, barriers, etc.) and/or bypass strategies (channel, levees, etc.). After reproducing the 25M event, we used our workflow to simulate different behaviours when artificially changing the fan river connectivity by means of a channel under different configurations of the debris flows surges. Our results show the importance of considering viscous and inertial debris flows to characterize the sediment transference dynamics in a tributary-junction alluvial fan.





## 2    Study Area and Data

In the *Huasco* Valley segments located between 1500 and 3000 $m.a.s.l.$, tributary-junction alluvial fans supply sediments to the main river and are so abundant that they can reach densities of 1.9 fans per $km$. During the 25M event, rainfall gauges in the valley recorded precipitation ranging from 20 to 76 $mm$ in three days, with a maximum intensity of 16 $mm$ $h^{-1}$. The high elevation of the snow line in March 2015 (3200 $m.a.s.l.$), 400 $m$ higher than its average altitude (Lagos and Jara, 2017), exposed areas usually covered by snow, thus increasing run-off (Wilcox et al., 2016; Jordan et al., 2019).

Many of the tributary catchments in the *Huasco* Valley were activated during the 25M event causing casualties and great economic losses (Izquierdo et al., 2021). The alluvial fans present a characteristic sedimentological response to the 25M event, where a sequence of surges impacted the fans repeatedly (Cabré et al., 2020a). The flood sequence of the 25M flood event was defined in the *Crucecita Alta* fan by Cabré et al. (2020a) because mitigation and reconstruction works were not performed immediately there, allowing a complete identification of the debris flow surges and geomorphologic evidence.

### 2.1    The *Crucecita Alta* fan

The studied fan is situated at the river junction of an ephemeral catchment with the main river ($\sim$29°S, 70°W) (Fig. 1.a). Its catchment has an area of $\sim$13 $km^2$, a length of 6 $km$, and a mean slope of 30%. The river junction is at 1044 $m.a.s.l$ and the maximum height of the catchment is 3129 $m.a.s.l.$; consequently, the whole catchment was under the snow line elevation during the 25M event. Catchment lithology is dominated by volcanic rocks (andesites), conglomerates, sandstones, and mudstones and by large accumulations of unconsolidated sediments in hillslopes and alluviated channels (Cabré et al., 2020a). The *Crucecita Alta* fan has 0.14 $km^2$ area; 8.6% slope; and hosts few houses in the northern area (Fig. 1.b). The northern area of the fan was not affected during the 25M event, presumably due to the presence of a 5 $m$ high deflection barrier made of unconsolidated debris. Riverbank erosion has previously trimmed the alluvial fan toe to the event (Cabré et al., 2020a), disrupting the alluvial fan gradient (Fig. 1.b). In *Crucecita Alta*, the difference in elevation between the main river and the alluvial fan surface is 13.5 $m$. This geomorphic configuration promotes the fan entrenchment and the formation of new fan lobes at its toe during debris flow events. These lobes may act as barriers because the valley is narrow with widths of 200-300 $m$.

A post-event topography of a 51 $km$ long segment of *El Carmen* River valley was acquired with $1 \times 1$ $m^2$ horizontal resolution between Feb-March 2017 by the Chilean Ministry of Public Works as part of a debris flow mitigation project in the area (IDIEM, 2019). The vegetation and buildings were removed from this available topography. Figure 1.a shows the river segment where the *Crucecita Alta* fan is located. In the pre-event satellite imagery, retrieved from ©Google Earth Pro v.7.3.3.7786 (CNES/Airbus images), there is no evidence of a feeder channel connecting the fan apex with the main river.

A calibrated HEC-HMS hydrological model (USACE, 2015) of the entire *El Carmen* River basin is also available from the same mitigation project (IDIEM, 2019; Zegers et al., 2020). Water flow discharge at the catchment outlet and at the river previous to the junction were obtained from this hydrological model. The obtained hydrograph shows that the flood event consisted of four main surges, with a maximum peak flow of $\sim$7 $m^3$ $s^{-1}$ during the fourth surge (Figure 2.b).


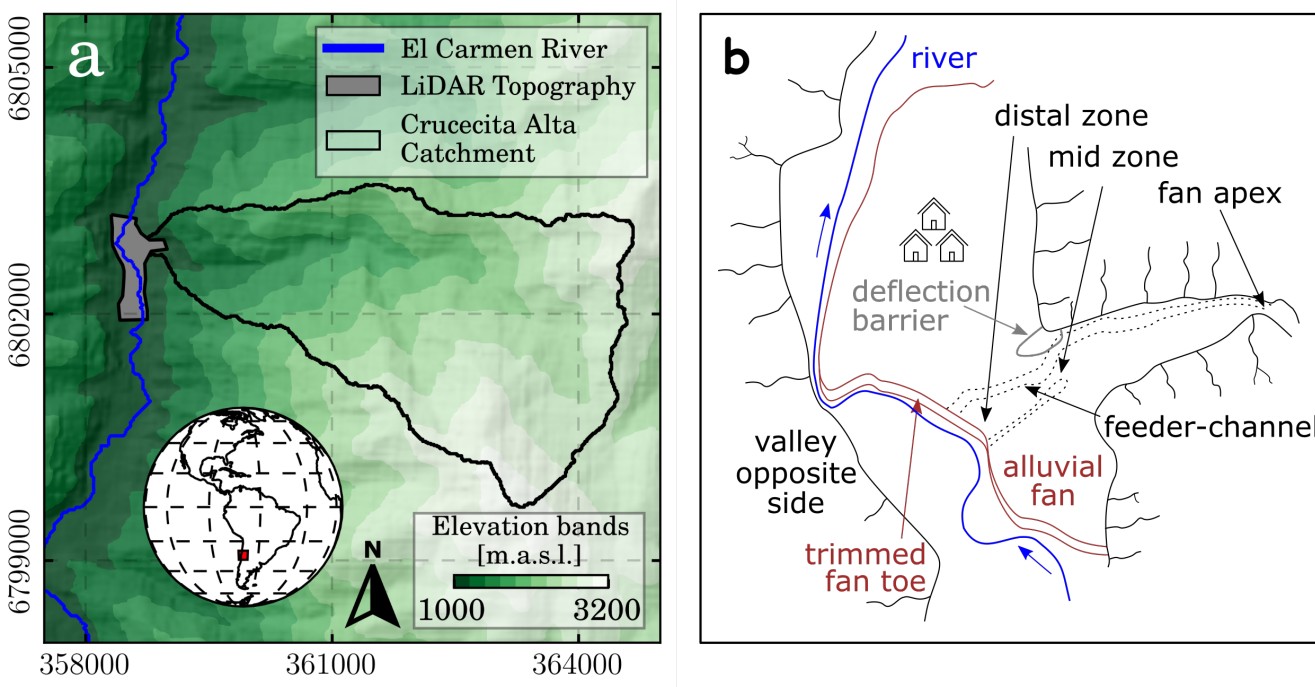

**Figure 1.** Study Area. (a) *Crucecita Alta* catchment (∼29°S, 70°W). The grey polygon is the river segment that includes the studied fan topography section used in the numerical model. The unfilled polygon depicts the *Crucecita Alta* catchment (13 $km^2$). (b) *Crucecita Alta* alluvial fan main geometric features where the feeder-channel generated during the 25M event are presented with dotted lines.

## 2.2 Characteristics of the 25M event

Cabré et al. (2020a) characterized the flood event sedimentology and spatial distribution of the five detected deposits in the
*Crucecita Alta* alluvial fan. These deposits were mapped and classified into facies (Fig. 2.a). The different facies were named
from $F1$ to $F5$, where $F1$ was the first deposit during the storm, and $F5$ was the last one. "Sedimentary facies" or "facies"
is a concept widely used in sedimentological studies because it assists in summarizing the grain shape, textural parameters,
the relief, and the stratification type onto a single area or zone that can be mapped (Wells and Harvey, 1987). As described
by Cabré et al. (2020a), $F1$ and $F2$ correspond to debris flow deposits associated with matrix-supported flows with high fine
sediment concentrations. $F1$ is the thickest deposit (above 1 $m$) and reaching near 7000 $m^3$ of deposited sediment volume
(Aguilar et al., 2020). Deposition occurred in a narrow area in the upper portion of the fan without reaching the main river. $F2$
is a wider but lower thickness deposit (10 - 30 $cm$) that overlays $F1$ and covers all the fan's length, from the apex to the distal
toe, with near 5000 $m^3$ of deposited sediment volume (Aguilar et al., 2020). $F1$ and $F2$ deposits overlain pre-event sediments,
indicating that the associated flows had none or negligible erosion capacity (Cabré et al., 2020a).
Following surges are responsible for the alluvial fan entrenchment and consequent generation of the feeder channel (Figure
2.b). These surges corresponded to more transitional flows and generated the $F3$, $F4$, and $F5$ deposits. $F3$ deposits are asso-


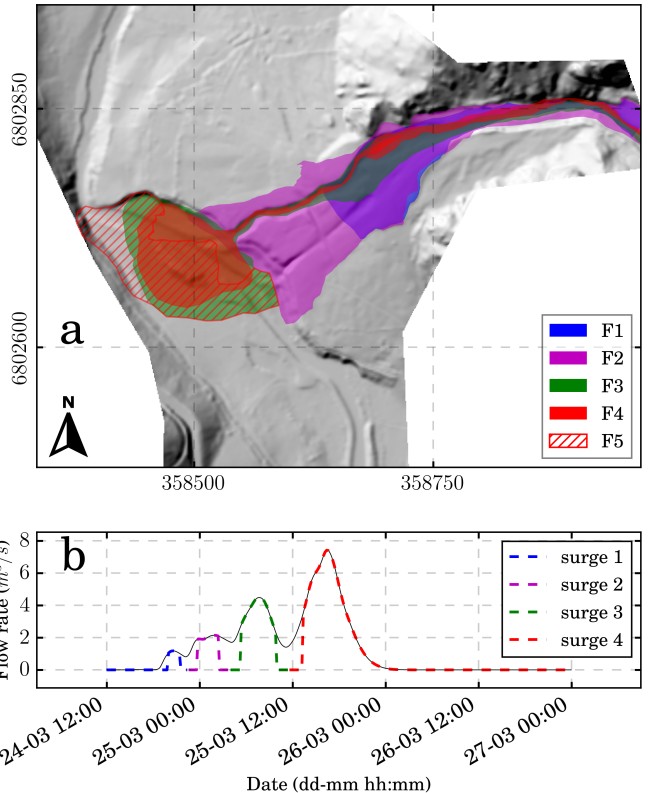

**Figure 2.** Available data for 25M event in *Crucecita Alta* alluvial fan. (a)The facies F1 (blue), F2 (magenta), F3 (green), F4 and F5 (red) retrieved from Cabré et al. (2020a) and LiDAR topography surveyed by IDIEM (2019). (b) Flow hydrograph obtained from the hydrologic model performed by IDIEM (2019). The colors used to identify the facies in (a) depict their correlation with the surges in (b).

ciated with hyper-concentrated and cohesionless transitional flows (Cabré et al., 2020a) (i.e., stony debris flows in Takahashi (2014)). These flows were channelized below the middle portion of the fan and formed the feeder channel. During the formation of $F3$, sediment deposition mainly occurs in the fan's upper part and after the fan toe in the *El Carmen* river. After this phase,

more dilute flows erode the previously deposited materials and enable the deposition of $F4$ and $F5$ facies further downstream. Feeder channel depth ranges from 100 to 350 $cm$ in the mid zone and reaches up to $\sim$500 $cm$ in the fan toe (distal zone). The local base level, controlled by the main river, was reached during the incision. Therefore, the subsequent flows were deposited in a new lobe at the fan toe, exhibiting a telescopic-like morphology with open framework boulders and lobate gravel lenses. These deposits correspond to inertial debris flows with clast supported fabrics and a matrix-free top (Cabré et al., 2020a).

In our analysis of the 25M event, the cross-cutting relationships of the facies mapped in *Crucecita Alta* by Cabré et al. (2020a) ($F1$ to $F5$) are fundamental evidence to determine the sequence of flows. Cabré et al. (2020a) interprets the significance of the mechanical classification of debris flows and classifies $F1$ and $F2$ as non-Newtonian flows. In contrast, $F4$ and $F5$



are classified as Newtonian flows, which is a similar differentiation, to some extent, to the viscous and inertial debris flows classification of Takahashi (2014). $F3$ is a transitional flow between the observed non-Newtonian and Newtonian flows.

## 3 Methodology

### 3.1 Governing equations

We use the two-dimensional FLO-2D model to solve the flood wave progression for water and debris flows in complex topographical terrains (O'Brien and García, 2009). FLO-2D solves two-dimensional depth-averaged continuity and momentum equations, Eq. (1) and Eq. (2), known as Saint–Venant equations, for water and debris flows.

$$\frac{\partial h}{\partial t} + \frac{\partial h V_x}{\partial x} + \frac{\partial h V_y}{\partial y} = 0 \tag{1}$$

$$\begin{aligned}
S_{f_x} &= S_{0_x} - \frac{\partial h}{\partial x} - \frac{V_x}{g}\frac{\partial V_x}{\partial x} - \frac{V_y}{g}\frac{\partial V_x}{\partial y} - \frac{1}{g}\frac{\partial V_x}{\partial t} \\
S_{f_y} &= S_{0_x} - \frac{\partial h}{\partial y} - \frac{V_x}{g}\frac{\partial V_y}{\partial x} - \frac{V_y}{g}\frac{\partial V_y}{\partial y} - \frac{1}{g}\frac{\partial V_y}{\partial t}
\end{aligned} \tag{2}$$

where $h$ is the flow height, $V_x$ and $V_y$ are flow velocities in the $x$ and $y$ directions and $S_{0_x}$ and $S_{0_y}$ are the channel slopes in each transverse direction $(x, y)$. $Sf_{x,y}$ denotes the frictional slope, accounting for flow resistance. In the case of water flows, $Sf_{x,y}$ is calculated using the Manning equation. In the case of debris flows, $Sf_{x,y}$ is calculated using the so-called quadratic rheology model, Eq. (3), which combines different terms accounting for yield/coulomb, viscous, and turbulent/dispersive stresses (O'Brien and García, 2009).

$$S_f = \frac{\tau_y}{\gamma_m h} + \frac{K\eta V}{8\gamma_m h^2} + \frac{n_{td}^2 V^2}{h^{4/3}} \tag{3}$$

where $\tau_y$ is the yield stress, $\gamma_m$ the specific weight of the solid-liquid mixture, $h$ the element (cell) flow depth, $K$ a resistance parameter for laminar flow, $\eta$ the dynamic viscosity of the fluid phase, $V$ the element (cell) flow velocity and $n_{td}$ a pseudo-Manning coefficient corrected by the sediment concentration. O'Brien and García (2009) suggest the following empiric relationships, Eq. (4), for estimating the parameters $\eta$, $\tau_y$, and $n_{td}$ as functions of sediment volume concentration, $C_V$, (O'Brien et al., 1993; O'Brien and García, 2009).

$$\begin{aligned}
\eta &= \alpha_1 e^{\beta_1 C_V} \\
\tau_y &= \alpha_2 e^{\beta_2 C_V} \\
n_{td} &= nb e^{m C_V}
\end{aligned} \tag{4}$$

where $\alpha_{1,2}$ and $\beta_{1,2}$ are empirical coefficients that have to be calibrated. $n$ is the conventional Manning coefficient, and $b = 0.0538$ and $m = 6.0896$.

The model also used a detention coefficient ($SD$) that controls flow detention. D'Agostino and Tecca (2006) suggest that $SD$ works as a minimum physically plausible flow depth. In addition, Zegers et al. (2020) reported that $SD$ is one of the



two more sensitive parameters in the model and, even though it is not part of the rheological model, it can surrogate the flow rheology.

For water flows, FLO-2D is also capable of solving sediment transport equations to simulate mobile bed and topographic adjustments. In the FLO-2D model, the sediment transport computation is uncoupled from flow hydraulics, i.e., water flow characteristics are computed first, and then sediment transport and bed geometry changes due to sediment erosion or deposition are computed for that timestep (O'Brien and García, 2009). However, the mudflow model assumes a fixed bed channel with no erosion/deposition process when using the mudflow model.

The sediment transport model accounts for the local deficit/excess of transported sediment, modifying the bed channel height using the well-known Exner, Eq. (5).

$$\frac{\partial h_0}{\partial t} + \frac{1}{1-p}\left(\frac{\partial q_{s_x}}{\partial x} + \frac{\partial q_{s_y}}{\partial y}\right) = 0 \tag{5}$$

where $h_0$ is the height of the channel bed, $q_s$ is the sediment load, and $p$ is the porosity. For the sediment load ($q_s$), FLO-2D has eleven different expressions based on unique river conditions (O'Brien and García, 2009). For this study, we used the

Parker-Klingeman-Mclean equation (Parker et al., 1982), Eq. (6). This equation is suitable for gravel and sandy bed material and simulates sediment transport mechanisms from bedload to mixed suspended sediment loads.

$$q_s = \frac{W^* u_*^3 \rho_s}{(s-1)g} \tag{6}$$

where $W^*$ is the dimensionless sediment transport rate, Eq. (7), $u_* = \sqrt{\tau_0/\rho}$ the frictional velocity, $\tau_0$ the bottom shear stress, $\rho_s$ the density of the sediments, $\rho$ the water density, $s = \rho_s/\rho$ the specific weight and $g$ the gravitational acceleration.

$$W^* = \begin{cases} 11.2\left(1 - \frac{0.822}{\phi_{50}}\right)^{4.5} & \phi_{50} > 1.65 \\ 0.0025\,\exp\left[14.2(\phi_{50}-1) - 9.28(\phi_{50}-1)^2\right] & 0.95 \le \phi_{50} \le 1.65 \\ 0.0025\,\phi_{50}^{14.2} & \phi_{50} < 0.95 \end{cases} \tag{7}$$

where $\phi_{50}$ is the normalized Shields stress, also known as transport stage, Eq. (8):

$$\phi_{50} = \frac{\tau_{50}^*}{\tau_{r_{50}}^*}$$
$$\tau_{50}^* = \frac{u_2^*}{(s-1)gD_{50}} \tag{8}$$

where $\tau_{50}^*$ is the Shield Stress, $\tau_{r_{50}}^* = 0.0876$ is a reference Shields stress value, and $D_{50}$ the mean sediment size of the substrate.

Originally, Eq. (7) considered $W^* = 0$ for $\phi_{50} < 0.95$, but it was modified by Parker (1990) to have a positive transport rate in every flow condition. Parker and Klingeman (1982) did another modification for multiple sediment sizes, where the dimensionless fractional transport rate $W_i^*$ for diameter $i$ is calculated as a function of Eq. (9).

$$\phi_i = \frac{\tau_{50}^*}{\tau_{r_{50}}^*}\left(\frac{D_{50}}{D_i}\right)^{0.018} \tag{9}$$


FLO2D model allows the user to specify the sediment gradation coefficient, $S_{CC}$, Eq. (10), which is a metric of the sediment size distribution.

$$S_{CC} = \frac{(D_{30})^2}{D_{10} D_{60}} \tag{10}$$

Cabré et al. (2020a) reported that viscous debris flows observed in the *Cucecita Alta* fan show negligible erosion. Therefore, the mudflow model of FLO-2D is enough to represent the flow characteristics of viscous debris flows. On the other hand, Cabré et al. (2020a) indicates that inertial debris flows observed in the *Cucecita Alta* fan significantly modified its morphology, so erosion and deposition processes can not be neglected. Since FLO2-D can not use the mudflow and sediment transport models simultaneously, we used the following reasoning. For viscous debris flows, the first two terms of Eq. (3), i.e., yield and viscous stresses, dominate flow resistance. Conversely, for inertial debris flows, the third term of Eq. (3), i.e., turbulent and dispersive stresses, becomes more important for estimating $S_f$. Because of this behavior, inertial debris flows can be modeled using the same Manning approach for water flows but considering an appropriate Manning coefficient. This approach has been validated in previous studies, where the Manning coefficient was estimated by back-calculation (Rickenmann et al., 2006; Naef et al., 2006). Typically used values for the Manning coefficient for modeling turbulent Newtonian debris flows are around 0.1 (Rickenmann, 1999).

## 3.2 Model configuration

The FLO-2D model built for this tributary-junction alluvial fan considers a 1400 $m$ long river section and a 450 $m$ long section of the alluvial fan, from the fan apex to the river junction (Fig. 3). To have an adequate flow development and achieve the balance of the sediment transport capacity before reaching the fan apex, the LiDAR topography of *Crucecita Alta* was extended 550 $m$ upstream using ALOS PALSAR DEM (12,5 x 12,50 $m^2$ resolution: https://search.asf.alaska.edu/#/?dataset=ALOS). Secondly, because the ALOS PALSAR topography does not have the resolution to represent the creek topography, a synthetic channel was created in the extended zone to better represents the channel morphology. The synthetic channel dimensions were mapped based on the Google Earth Pro scenes. Thus, the model is 1 $km$ long from the *Crucecita Alta* catchment (from the inflow element to the river junction), where 450 $m$ corresponds to the LiDAR topography and 550 $m$ to the synthetic channel (Fig. 3). No pre-event high-resolution topography is available for the *Crucecita Alta* fan. This is a common problem when studying the geomorphic consequences of low-frequency, high-magnitude debris flow events in arid areas (Zegers et al., 2017). To overcome this limitation, similarly to the approaches used by McMillan and Schoenbohm (2020), the post-event raster was modified to reproduce the pre-event topography based on pre-event available imagery and field evidence. The pre-event geomorphological configuration of the *Crucecita Alta* fan (Fig. 3) consists of an undisturbed fan surface without evidence of erosion (caused by the incision of a feeder channel or headward erosion). The modifications performed over the post-event LiDAR topography to obtained the pre-event geometry can be summarized into: (i) The incisions caused by the 25M flood event in the alluvial fan were removed from the DEM, and then the surface was smoothed, (ii) facies F1 and F2 were subtracted of the DEM based on field thickness measurements of the deposits and assuming that cohesive debris flows have little to negligible bed erosion (Cabré et al., 2020a).




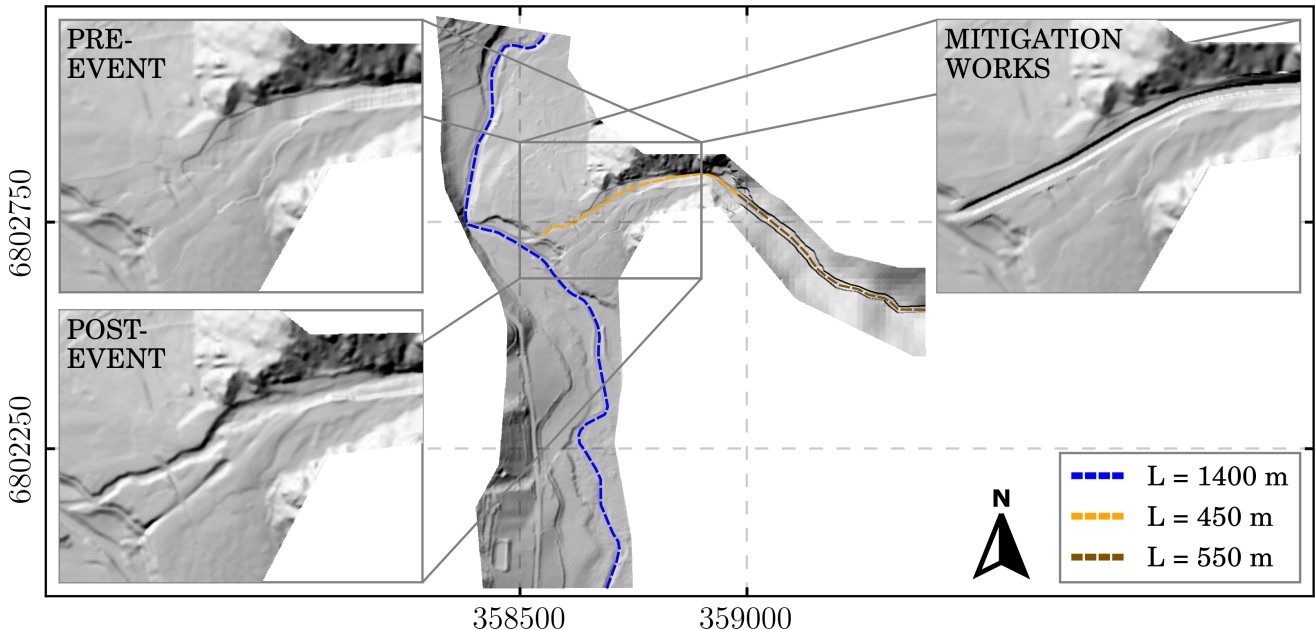

**Figure 3.** Topographic data modifications. Post-event topography corresponds to the available LiDAR topography while the pre-event topography is a restitution based on satellite images and the available topography. The synthetic channel (brown dashed line) attached to the LiDAR topography has a longitude of 550 $m$ while feeder channel (orange dashed line) has a longitude of 450 $m$. In the post-event topography, the feeder channel is the result of the inertial debris flows incisions on the alluvial fan. In the mitigation works topography, the feeder channel was replaced by a straight rectangular channel.

We also study the effects that a bypass channel, a typical debris flow mitigation work in alluvial fans, has on the fan-river connectivity. To this purpose, a ten meter wide and ten meter depth rectangular channel was inserted in the pre-event topography (Fig. 3).

The resolution of the numerical grid was set at 5x5 $m^2$, as a finer resolution results in numerical instabilities due to the high flow velocities. In contrast, a coarser grid resolution results in a loss of terrain information. Manning number, $n$, was set equal to 0.1 $s\ m^{-1/3}$ for the alluvial fan, which is within the range suggested by Rickenmann et al. (2006) (0.07–0.16 $s\ m^{-1/3}$) for debris flows. Although the incisions measured on the field reaches up to 5 $m$ deep, erosion depth was limited to 4 $m$, as greater depths affect the model stability. No sediment rating curve was set for the inlet because the 550 $m$ synthetic channel allows the model to find its own sediment load equilibrium before reaching the fan apex.

### 3.3 Modeling approach and workflow

The flood sequence of the 25M event consisted of four different debris flow surges with different sediment loads. The differences in the rheology of the flows resulted in multiple geomorphological adjustments within the fan. Viscous debris flows





mainly deposited sediments in the fan surface. In contrast, inertial debris flows strongly incised and eroded the fan. Conse-

quently, our modeling approach divides the debris flow event between viscous and inertial debris flow surges and correlate them to the field evidence (Fig. 2). Therefore, if the facies (or any other sedimentological description) indicates that a viscous debris flow formed the deposit, the associated surge is modeled with the mudflow model. Conversely, if the facies suggests an inertial debris flow, the surge is computed using the water flows model with $n =0.1$ and including sediment transport. For this purpose, we developed a python routine to concatenate the four surges into one model. Each surge is modeled in a sequence

where the results of the previous surge modify the topography for the next surge. The workflow steps are presented in Fig. 4.

In the first step of this workflow, "Subdivision into surges" (Fig. 4), we set the number of surges $i = 1...N$ and the rheology model of each one, i.e., non-Newtonian (viscous debris flow) or Newtonian (inertial debris flow). For example, facies $F1$ and $F2$ have high fine sediment concentration, present evidence of a laminar flow, and show negligible erosion. Therefore, surges 1 and 2 are assigned as viscous debris flows. In contrast, the following surges 3 and 4 are assigned as inertial debris flows since

$F3$, $F4$, and $F5$ show significant erosion and deposition and consist primarily of coarse sediment.

For the mudflow branch of the workflow (Fig. 4)) the next step is the "Calibration process and result screening". This step finds the best set of rheological parameters, as explained in the next section. Multiple model runs may pass the screening. Therefore, in the step "Model selection", the user must visually inspect all the filtered runs and choose the best fit for the characterized facies. This step could not be automated because a critical analysis is needed. From the selected run, the deposit

depth $h_d$ is estimated for each grid element according to equation 11, where $h_f$ is the resulting final flow depth reported by the numerical model and $C_V^{mean}$ is the mean volumetric concentration of the surge. The sediment rating curve function $C_V(Q,t)$ is presented in Zegers et al. (2020) and porosity $p$ is set equal to 0.3 for facies $F1$ and $F2$ (Nicolleti and Sorriso-Valvo, 1991). Finally, $h_d$ is added to the original topography at each cell by the python routine, which updates the model for the next surge.

$$h_d = h_f \frac{C_V^{mean}}{1-p}$$

(11)

$$C_V^{mean} = \frac{\int_{t=0}^{T} Q(t)C_V(Q,t)dt}{\int_{t=0}^{T} Q(t)dt}$$

On the other hand, the model is directly run for the sediment transport model branch of the workflow (Fig. 4) since no parameters need to be calibrated. The sediment transport model only needs the $D_{50} =7.8\ mm$ and $S_{CC} =1.79$ of the substrate of the *Crucecita Alta* alluvial fan (Cabré et al., 2020a). The sediment transport equation and the grain size distribution of the substrate do not change between surges. The sediment transport model of FLO-2D creates a file with the final bed elevation changes. The routine reads this file and modifies the next model data to update the topography for the next surge.

**3.4 Rheology calibration**

Zegers et al. (2020) performed a sensitivity analysis for the FLO-2D numerical model and established the most sensitive parameters in the mudflow model. In their study, two alluvial fans located in the same valley were tested. They concluded that equifinality was present in the model, mainly related to the rheological parameters. Thus, to reduce over parameterization and




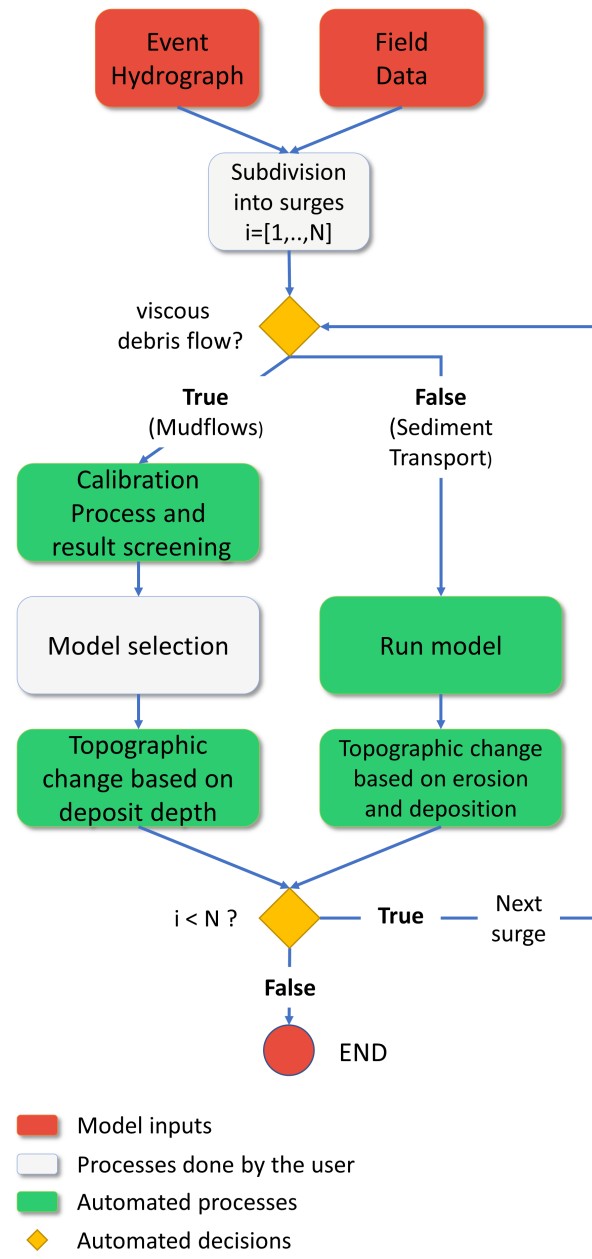

**Figure 4.** Surges modelling workflow. The main challenge is to find a correlation between surges and field data that allows the user to discriminate between viscous and inertial debris flow surges. The developed routine is able to concatenate the surges by updating the model inputs for the next surge, based on the results of the previous one.

restrict the parameter search, three parameters ($\alpha_1$, $\alpha_2$, and $SD$) have been left out of the calibration process. Consequently, the

calibrated parameter set correspond to the parameters $\beta_1$, $\beta_2$, $C_V^{max}$, and $Vol_{sediment}$ (i.e., a reduction from 7 to 4 parameters





in the calibration process). Parameters $\alpha_1$ and $\alpha_2$ were fixed since they are the least sensitive parameters of the model (Zegers et al., 2020) and were set equal to 0.0075 $poises$ and 0.152 $dynes\ cm^{-2}$, respectively, using the values obtained by Zegers et al. (2017). Conversely, the $SD$ parameter has a considerable influence on the results (Zegers et al., 2020). This parameter sets a minimum physically plausible flow depth, allowing the mixture to stop on the alluvial fan (i.e., the mudflow stops if the

local flow depth is lower than $SD$). However, based on estimated deposit height measurements by Cabré et al. (2020a), it was set equal to one meter. $SD$ was set as $1\ m$ for the alluvial fan and $0.03\ m$ for the river valley, where the value in the valley is the default value in FLO 2D for water flows.

We used the algorithm of Zegers et al. (2020) with slight modifications to create a calibration routine that allows us to run the model $N$ times and automatically apply screenings to determine the model's best fit. Similar to Mergili et al. (2018), we

assigned the pixels within the observed deposit as observed positives ($OP$) and pixels outside the deposit area as observed negative ($ON$). The pixels flooded by the simulation were assigned as predicted positive ($PP$), whereas the not flooded pixels as predicted negatives ($PN$). Hence, The screenings filter the results based on thresholds $\theta_1$ and $\theta_2$ associated with true positives ($TP= OP \wedge PP$) and false positives ($FP=ON \wedge PP$) (eq. 12). These thresholds were set in order to obtain less than five runs (out of one hundred) per surge, where $\theta_1 =0.7$ and $\theta_2=0.3$ for surge 1 and $\theta_1 =0.9$ and $\theta_2 =0.1$ for surge 2.

$$\frac{\sum px_i^{TP}}{\sum px_i^{OP}} > \theta_1 \qquad \frac{\sum px_i^{FP}}{\sum px_i^{OP}} < \theta_2 \qquad\qquad\qquad (12)$$

Zegers et al. (2020) compiled a wide range of physically plausible parameters from many studies in the literature. For the parameter sampling under these parameter ranges, we used the Latin hypercube sampling (LHS) method (Olsson and Sandberg, 2002). Since we simulated surges 1 and 2 with the mudflow model, both surges needed a calibration process to find their rheological parameters. We tested the model with sets of 50, 100, and 200 runs. Due to the constraint of calibrated parameters from seven to four, 100 model runs were sufficient to find reasonable results that fit the mapped deposits. For

50 runs, no result fits the screenings, whereas 200 runs show no significant improvements. The screenings applied, Eq. (12), returned four cases for each surge which were visually inspected to select the best fit. Considering the modified topography after the selected case for F1, we ran the same algorithm for the second surge. The minimum and maximum values for the calibrated parameters and the parameters adopted for surges 1 and 2 after the calibration process are presented in Table 1.

## 4   Results

In this section, we analyze if the proposed workflow can reproduce the debris flow event and the geomorphological adjustments that this alluvial fan experienced. Once the workflow capabilities are proven, we include a rectangular channel that connects the alluvial fan apex with the river. This channel is tested for the same chain of debris flow surges as in the 25M event and also for two inertial debris flow surges. Then, we study the effect of this channel on the lateral and longitudinal coupling status of

the river junction.




**Table 1.** Calibrated parameters and their range for surges 1 and 2. Parameters for both surges where expected to be different because of the equifinality and their different sedimentologic characteristics.

|  | $\beta_1$ | $\beta_2$ | $C_V^{max}$ | F1/F2 $Vol_{sediment}$ $[m^3]$ |
|---|---|---|---|---|
| min | 6.00 | 17.00 | 0.45 | 4000/3000 |
| max | 33.00 | 30.00 | 0.60 | 10000/7000 |
| Surge 1 | 20.23 | 17.20 | 0.58 | 9482 |
| Surge 2 | 18.14 | 20.55 | 0.50 | 4167 |

### 4.1    25M debris flow event reconstruction

The maximum and final flow depths of the viscous debris flow surges 1 and 2 are presented in the flood inundation maps of Fig. 5. In a first visual inspection of Fig. 5, the simulated flooded areas are in good agreement with the facies mapped. Integrating the inflow hydrograph and considering the variable $C_V$, for surge 1, the inflow volume is 15450 $m^3$ of water and 16991 $m^3$ of sediment resulting in $C_V^{mean}$= 0.52. According to the simulation, the volume of sediment deposited on the alluvial fan is 9482 $m^3$ (Fig. 5.a and 5.b). In contrast, the volume estimated from field measurements is ca. 7000 $m^3$ (F1). The integration of the inflow hydrograph for surge 2, on the other hand, results in an inflow volume of 28218 $m^3$ of water and 10453 $m^3$ of sediment, i.e., $C_V^{mean}$=0.27. The lower value for $C_V^{mean}$ for surge 2 indicates a lower flow resistance than surge 1. According to the simulation, the deposited sediment volume is 4167 $m^3$ (Fig. 5.c and 5.d), whereas the volume estimated from field measurements is 5000 $m^3$ (F2).

Surge 1 does not reach the river due to its high $C_V^{mean}$ and the absence of a channel (Fig. 5.b). In a fan without a feeder channel, viscous flows spread over the surface, increasing flow resistance, promoting flow detention, and consequent sediment deposition (Jakob and Hungr, 2005). Surge 2 reaches the river (Fig. 5.d), probably given to its more diluted nature (lower $C_V^{mean}$ than surge 1). Thus, a small portion of the sediment transported by surge 2 reaches the river, but it was insufficient to change the river geometry. Therefore, the alluvial fan acts as a sediment buffer, preventing the interaction with the main river. Consequently, the river's sediment discharge remains almost undisturbed even though the occurrence of surges 1 and 2.

The $SD$ parameter controls the flow depth at the end of the simulation for surge 2 (Fig. 5.d). Since $SD$ =1 $m$ on the alluvial fan, almost every cell remains with a fixed water depth of 1 $m$ at the simulation's end. We expected this condition because the simulated deposited volume is very sensitive to $SD$ when the alluvial fan operates as a buffer preventing sediments from reaching the river (Zegers et al., 2020). However, $SD = 1\ m$ overestimates the sediment deposit thickness. We also tested lower $SD$ values 0.5 $m$ and 0.7 $m$, associated with deposit thicknesses of 0.2 $m$ and 0.3 $m$ according to equation 11, obtaining unsatisfactory results regarding the observed affected area. The problem with lower $SD$ values is that the flow spreads over the



**Figure 5.** Viscous debris flow surges. Top panels correspond to the results for surge 1 whereas bottom panels correspond to the results for surge 2. Dashed polygons show the extent of mapped facies F1 (blue) and F2 (red). Left panels present the maximum flow depth whereas right panels present the final flow depth at the end of each surge. Topographic changes for subsequent surges are updated based on the final flow depth of the previous surge.

surface, losing any correlation with the mapped facies. Therefore, we prioritize matching the flooded area instead of the deposit thickness for the model calibration, as measuring the flooded area is more reliable than characterizing the deposit thickness

distribution.

Our simulation results for the following inertial debris flow surges 3 and 4 are presented in Fig. 6. These surges have a higher amount of water than the previous viscous surges. Surge 3 has a volume of water of 61388 $m^3$ and surge 4 of 143568 $m^3$.





The maximum flow depth maps of Fig. 6.a and Fig. 6.c evidence the river avulsion triggered by surge 3, where the original river path and the newly formed river path are observed. The river is pushed to the opposite valley side due to surge 4 with

flow depths over $2\ m$ deep. The resulting topographic changes for surges 3 and 4 (Fig. 6.b and Fig. 6.d) are coherent with the primary erosion and deposition zones identified during fieldwork. Moreover, it is possible to locate the new lobe formed at the fan toe showing a telescopic-like pattern responsible for the river avulsion and the partial river blockage. On the other hand, the new scoured channel on the alluvial fan is wider than the channel observed on the field, possibly due to the numerical model resolution. These results show that the fan-river interactions become important for surges 3 and 4 because the tributary

sediment discharge experiences a complete coupling with the main river through alluvial fan trenching and progradation. Savi et al. (2020) reported similar behavior in their laboratory experiments.

We also analyzed the potential river blockage or dam formation. The obstruction ratio represents the space occupied by sediment deposits compared to the available space for the river to flow. (Stancanelli and Musumeci, 2018) This ratio is useful to discriminate between three states (no blockage, partial blockage, full blockage). The obstruction ratio $r_b$ is defined as the

ratio between the obstruction lobe width and the flooded valley width, both along the orthogonal direction to the river flow (Stancanelli and Musumeci, 2018). The value $r_b = 1$ means a complete river blockage. River blockage or dam formation is dominated by the momentum ratio $R_M$ and the unevenness of the grain sizes $S_C$ (Dang et al., 2009). The momentum ratio is calculated as the product of the flow rate ratio $R_Q$, the velocity ratio $R_V$, the bulk density ratio $R_\gamma$ and the confluent angle $\theta$ while $S_C = \sqrt{D_{75}/D_{25}}$. Dang et al. (2009) proposed a critical index $C = R_M S_C$ for dam formation with different partial

and complete blockage thresholds. Stancanelli and Musumeci (2018) highlighted that the thresholds have to be calibrated depending on the material adopted. For gravel deposits, they proposed a threshold value of $C = 9$.

For surges 3 and 4, we obtained $C$ indexes of 2.15 and 0.85, respectively. These values indicate that both surges do not have the potential to block the river. The $C$ indexes are consistent with the obstruction ratios $r_b$ of 0.75 (surge 3) and 0.86 (surge 4). These $r_b$ values indicate a river's partial blockage due to the lateral input of sediment. Interestingly, the river experiences

deposition in its downstream section for surge 3, i.e., an excess of sediment load (Fig. 6.b). Conversely, the river experiences erosion in the downstream section for surge 4, i.e., a deficit of sediment load (Fig. 6.d). This change indicates that a river obstruction $r_b = 0.75$ was not able to act as a barrier. In contrast, an obstruction ratio $r_b = 0.86$ was able to act as a barrier decoupling the river's sediment discharge in the river junction. For both inertial surges, the upper section of the river is a deposition zone because of the river's partial obstruction and, consequently, pounded water.

**4.2   Morphological evolution of the tributary-junction alluvial fan**

We characterized the tributary-junction alluvial fan evolution by analyzing the topographic change using four longitudinal sections (Fig. 7.a). All longitudinal profiles are presented in the downstream direction. We established the A-A' profile of the alluvial fan according to the main incision observed in the field after the 25M event. In contrast, sections B-B', C-C', and D-D' correspond to the main river, which shifted during the event.

River avulsion is well defined in profile A-A' (Fig. 7.b), where the river channel shifts to the opposite valley side after surges 3 and 4 due to the formation of a new lobe. Profile A-A' shows a convex shape after surge 1, typical for viscous debris



**Figure 6.** Inertial debris flow surges. Top panels correspond to the results for surge 3 whereas bottom panels correspond to the results for surge 4. Dashed polygons delimit the telescopic-like deposit, i.e. the incision in the alluvial fan and the new lobe at the fan toe. Left panels present the maximum flow depth while right panels present the topographic change for erosion (negative values) and deposition (positive values).

flows with sharp frontal boundaries. The following surge 2 stacks over surge 1, maintaining this convex shape and raising the topography. On the other hand, inertial debris flow surges 3 and 4 result in a characteristic concave shape for profile A-A' due to an alluvial fan incision. Analog laboratory experiments of Savi et al. (2020) reproduced similar sediment mobilization and

geomorphic changes. The results of our simulation show less scour than the observed incision on the field, possibly because of the fixed 4 $m$ of maximum erosion. Another explanation for the lack of erosion is the difference between eroding the early-





event deposits ($F1$ and $F2$) and the pre-event old debris flow deposits. In our routine, the new deposits are accounted as part of the topography instantly when our routine updates the topography between surges. However, the early-event sediments should be eroded easier than pre-event old deposits.

The B-B' profile follows the river channel observed in the LiDAR topography. Profile C-C' corresponds to the main flow path of the river after the new lobe associated with surge 3. Profile D-D' corresponds to the main final flow path at the end of surge 4 and, therefore, at the model's end. The upper and lower sections are the same for these three profiles, whereas the lobe section follows the river avulsion path. Therefore, profiles B-B', C-C', and D-D' are presented for $T_2$, $T_3$, and $T_4$, which correspond to the status of the main channel after surge 2, 3, and 4, respectively (Fig. 7.c). Initial status and the status of the

river after surge 1 remain the same as $T_2$ and are not presented here. Profile B-B' illustrates how the sediment yielded from the tributary reaches the river forming the new lobe of the telescopic-like deposit with max deposited depths around 6 $m$ for $T_3$ and 4 $m$ for $T_4$. Surge 3 is responsible for the first channel avulsion, represented in the C-C' profile for $T_3$, and evidences the new lobe extent. Surge 4 increases the fan progradation, so profile C-C' exhibits deposition for $T_4$ in the lobe section. During surge 4, the river shifts again, and the D-D' profile follows the final river path where $T_4$ evidence the erosion and formation of

the final river path.

### 4.3    Simulation of different scenarios considering mitigation works

In hazard assessment projects, numerical models are also used to test mitigation works. However, these models do not always consider the broad debris flow types that a creek can experience. Moreover, the same hydraulic works should be studied under different scenarios. As an example, we tested a rectangular channel (10m width and 10m depth) that connects the alluvial fan

apex with the river junction under two different scenarios: (1) Scenario 1 consists of the same sequence of debris flow surges observed in 25M event, where two viscous debris flows are followed by 2 inertial debris flows. (2) Scenario 2 consists only of the two inertial debris flows. With both scenarios, we prove the importance of studying different debris flow types combinations for the same mitigation work.

### 4.4    Scenario 1

In Scenario 1, the proposed channel confines viscous debris flows, which allows the flow to reach the river for surges 1 and 2. For surge 1 shown in Figure 8.a, deposit depth in the channel is up to 7 $m$ and over the river junction up to 4 $m$, indicating a partial channel obstruction. Surge 2, on the other hand, spreads over the river junction but does not form a new lobe. Surges 3 and 4, shown in Figures 8.c and 8d, are deviated and forced to flow in the upstream direction of the river due to the previous deposits. For surge 4, the channel overflows and inundates the fan. We first tested channels with smaller cross-sections where

overflow also occurred, whereas a larger channel cross-section would be cost-inefficient and, therefore, not realistic.

This case shows that a channel is not sufficient to convey the debris flows safely. However, deposits observed in the river junction are smaller than deposits observed in the original case. Surprisingly, the presence of a channel does not directly mean an increase in lateral connectivity. In the 25M event, alluvial fan trenching creates a local increase in the sediment load, which is immediately deposited at the fan toe due to the change of the slope. From this test, we learned two important features when


**Figure 7.** Morphological evolution of the tributary-junction alluvial fan. (a) Location of the longitudinal sections presented in b and c. (b) Longitudinal profile A-A' of the alluvial fan topography after each surge. (c) Longitudinal profiles B-B', C-C', D-D' of the river's topographic evolution.

assessing this debris flow hazard. First, trimmed fan toes create a base level drop that increases the sediment load that reaches the river, enhancing river avulsion. Second, transport-limited catchments such as *Crucecta Alta* produce sediment discharges



during extreme storm events, resulting in cost-inefficient works, which are not realistic. In Chile, what is being built are mitigation works that reduce the impact of debris flows but do not entirely solve the problem. Therefore, targeted population education to avoid people settling in areas under risk must join these works.

## 4.5  Scenario 2

Scenario 2 (Figures 8.e and 8.f) also demonstrate that the presence of a channel avoids the telescopic-like deposit. The channel avoids an abrupt base level drop due to the trimmed fan toe and, therefore, avoids headward erosion and the local increase of sediment load. Consequently, no local sediment load increase takes place in the alluvial fan for surges 3 and 4. For this scenario, only a mild river avulsion without river obstruction occurs for surges 3 and surge 4.

## 5  Discussion

### 5.1  Dynamic response of the fan-river connectivity

The field "forensic" analysis of the debris flow event in the *Crucecita Alta* fan resulted in a clear differentiation of the transport mechanisms for each surge (Cabré et al., 2020a). Our workflow benefits from this differentiation to select a specific model to represent the flow of the water-sediment mixture in each surge. Our results show that the combination of these numerical 420 models reproduces the main processes described by Cabré et al. (2020a). For the 25M event in *Crucecita Alta* alluvial fan, the coupling conditions changed within the event, but our workflow can reproduce this dynamic response. For the 25M event in the *Huasco* basin, Cabré et al. (2020a) reported 49 catchments with a characteristic response where seven catchments are described in detail. This specific response of the catchments consists of viscous debris flow surges are followed by inertial debris flow surges. This catchment response in the Atacama Desert also occurred in 2017 and 2020 but, unfortunately, has not 425 been reported.

Characteristic stratigraphic and geomorphic features observed in the *Huasco* river basin can also be used to understand changes in the water-sediment ratio and the coupling degree between fans and the main river. We observed on the field that sediment sources influence fine sediment vs. coarse sediment ratio present in the debris flows. On the one hand, viscous debris flows travel short distances because of their high flow resistance. Therefore, these surges come from sources close to the fan 430 apex and arrive first. Conversely, inertial debris flows can travel longer distances and come from more distant sources (i.e., they arrive later at the alluvial fan). We expect this behavior to keep occurring in these arid zones; therefore, our workflow could be used to study future scenarios. Moreover, our workflow could assist in the volumes estimations of water and sediment needed to modify the sediment cascade dynamics under viscous and inertial debris flows, which are different, as seen in this study.

Using our workflow on a past event in *Crucecita Alta*, we determined the following characteristics. The fan can completely 435 buffer the viscous debris flow surge 1 ($C_V^{mean} = 0.52$ and a volume of sediment around 9500 $m^3$). Conversely, for surge 2, a still viscous but more diluted debris flow, the fan partially buffers the sediment discharge ($C_V^{mean} = 0.27$ and a volume of sediment around 4200 $m^3$). Moreover, for both later inertial debris flows, the sediment discharge of the tributary is totally

**Figure 8.** Two future scenarios are modelled to understand the possible effects of a channel (mitigation works): First scenario (a,b,c,d) and second scenario (e,f).





coupled. Alluvial fan trenching for surge 3 increases the river's sediment load downstream, leading to total connectivity of the sediment cascade. Conversely, for surge 4, the greater volume of sediment acts as a barrier that decouples the sediment

discharge of the river.

We also studied how the *Crucecita Alta* alluvial fan would react to a channel. This approach has proven helpful because the channel (mitigation work) was expected to reduce affected areas on the alluvial fan and enhance the trunk river obstruction. However, the results in scenario 1 indicate that a channel reduces affected areas and avoids headward erosions. Thus, it reduces the amount of sediment that finally reaches the river reducing the river obstruction. In scenario 2, even though the affected

areas are reduced, deposition occurs between the apex and the fan's mid zone and, consequently, a channel overflow. However, headward erosion is also avoided for scenario 2, and the new lobe at the fan toe reduces its size considerably.

## 5.2 Model limitations

Even though our workflow reproduces the main processes of interest successfully, we observed two main limitations. First, we recognize that neglecting bank erosion may affect the sediment cascade dynamics. However, this process is rarely considered

because most numerical models do not account for bank erosion. Second, the Surface Detention parameter ($SD$) directly impacts on the deposited volume of sediment ($Vol_{sediment}$) in the calibration process because it controls the final flow depth. Therefore, it is advisable to estimate the $SD$ parameter based on field data properly. If no data of the deposit depth is available to estimate this parameter, $SD$ could be added to the calibration process. Here we calibrated a parameter set of 4 unknowns ($\beta_1, \beta_2, C_V, Vol_{sediment}$) whereas a calibration procedure with 5 unknowns increases exponentially the computational costs.

In the absence of a proper estimation of $SD$, we recommend that the user chooses a value between 0.5 and 1 $m$ for viscous debris flows in steep alluvial fans in the Atacama Desert instead of adding $SD$ to the calibration process. This will maintain computational costs reasonably. The range $[0.5, 1]$ $m$ is based on this study's findings and other previous studies in the Atacama Desert, such as Zegers et al. (2017, 2020). In spite of these limitations, it is shown here that fan-river interaction studies can be performed with a commercially available software that simplifies the physics around the flow of solid-liquid mixtures.

## 5.3 Insights for a new modelling approach

Low-frequency, high-intensity rainstorm events in the Atacama Desert have increased their occurrence in the last century, and it is expected that this trend continues due to climate change (Ortega et al., 2019). A higher frequency in high-intensity rain events in these transport-limited (i.e., sediment-unlimited) catchments (Aguilar et al., 2020) result in a higher frequency in the debris flows events. Following this reasoning, the necessity of properly reproducing the debris flows mechanics should be a priority.

Most of the models focus on run-out distance and affected area, but none cope with the notorious changes of rheological macroscopic behaviors between surges. Based on the data for a specific event, our workflow studies the tributary-junction alluvial fan's response to a characteristic chain of flows with different rheological behaviors in the arid region of Chile (27°S - 30°S). This study highlights the effects of an alluvial fan on the lateral and longitudinal connectivity of the sediment cascade. Our workflow can be used to study future scenarios where the catchment has a specific sedimentological response to rainfall

events. Moreover, we show the decisive and dynamic role of an alluvial fan in the tributary-river junction (dis)connectivity





and, consequently, in the transference of the sediment down-system. To our knowledge, this important interaction is not yet considered in hazard assessment projects but impacts considerably in the sediment budget analysis of a fluvial system. This work has presented new insights on how debris flow hazards should be simulated in a tributary-junction alluvial fan. In the future, we expect to propose a modeling protocol to guide modelers on how to study these important but neglected effects on catchments in northern Chile.

## 6 Conclusions

This study addresses how a tributary-junction alluvial fan results in local perturbations over the sediment cascade dynamics. We created a novel methodology based on the conceptual models of Mather et al. (2017); Savi et al. (2020); Cabré et al. (2020a). The proposed workflow reproduces sequential mass flow surges that reach the *Crucecita Alta* fan and its morphologic adjustments. Our approach copes with the challenge of coupling different simulation models by integrating cascading mass flow processes in one integrated modeling workflow. In *Crucecita Alta* alluvial fan, we first reproduced the fan aggradation and the tributary's sediment discharge buffering for viscous debris flow surges 1 and 2 (Fig. 5). These surges do not impact the river flow, and the effects over the sediment cascade are negligible due to the high rates of deposited sediment in the alluvial fan. Surges 1 and 2 are therefore uncoupled from the river's sediment discharge. Second, inertial debris flow surges 3, and 4 incise and prograde the alluvial fan. Although the zone near the apex may still be aggrading, the mid-zone gets incised, creating a local increase of sediment load that reaches the river and deposits forming a new lobe. The newly formed lobe evidences the coupled status of the tributary with the river. Furthermore, the narrow shape of the valley enhances river blockage that decouples the river's upper and lower sections (Fig. 6). In conclusion, coupling conditions for the tributary-junction alluvial fan are dominated by the viscous/inertial debris flow nature due to their capacity to travel shorter/longer distances, respectively. This study depicted the significance of the mechanical classification of debris flows from an engineering perspective to reproduce how an alluvial fan controls the tributary-river junction connectivity and the effects of building a channel as mitigation work.

Fan-river interactions are a key process for hazard assessment because they directly impact the sediment transfer dynamics and the landscape evolution modeling. Usually, these interactions have been studied from a geological point of view, but they are not considered from an engineering perspective in numerical models. No standard procedure exists in hazard assessment projects involving the numerical simulation of debris flows to account for these fan-river interactions. Our methodology provides new insights into how to improve these engineering studies in these arid systems, such as the Atacama Desert, where the catchments have a characteristic sedimentological response (set of facies) to an extreme storm event. Studies considering only the mudflow model, i.e., not taking into account erosion/deposition processes, result in underestimated affected areas due to sediment load changes and the avulsion of the river. On the other hand, studies considering only the sediment transport model in water flows do not consider the effects of non-Newtonian flows such as channel plugs due to highly viscous debris flows. The complex combination of different debris flow types makes it impossible to know beforehand how the fan and the mitigation works will behave, so the best approach is to study the solution under different scenarios by combining viscous and inertial debris flow surges. Our methodology is a good solution to represent debris flows and their effects on the



tributary-junction alluvial fan connectivity, taking advantage of the capabilities of existing one-phase models. However, nu-

505 merical models should better resolve erosion and deposition processes and changing rheologies to represent sediment transport

processes in tributary-junction alluvial fans.

*Author contributions.* A.G., G.Z., A.C., G.A., and S.M. were involved in the conceptualization, methodology, analysis, original draft manuscript. A.G. and G.Z developed the model. A.G. conducted the simulations, validation and made the figures. A.G., G.Z., A.C., G.A., A.T., and S.M contribute to writing, review & editing. SM project administration.

510 *Competing interests.* The authors declare that they have no conflict of interest.

*Acknowledgements.* This research was supported by Advanced Mining Technology Center (AMTC) and the Department of Civil Engineering from Universidad de Chile; the National Agency of Research and Development (ANID); ANID/PIA Project AFB180004. AG and GZ were financially supported by the ANID Doctoral grants 21191593 and 72200390. The authors thank the Public Works Ministry (DOH-MOP) for the LiDAR topographical data.



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
