# Peer review of "A modeling methodology to study the tributary-junction alluvial fan connectivity during a debris flow event"

_Natural Hazards and Earth System Sciences, 2021_

## Referee Comment (RC2)

**A review of:**
**A modeling methodology to study the tributary-junction alluvial fan connectivity during a debris flow event**

Eric A. Barefoot

October 27, 2021

**Synopsis**

Garcés et al. present a modeling study of an alluvial fan in southern Chile, which suffered a series of debris flows in 2015. The objective of the modeling is to use a reduced complexity approach that captures the broad patterns of sediment dispersal without having to explicitly model the complex physics of debris flows. In this way, different mitigation strategies (e.g. basins, spillways) can be tested without extreme computational expense. The authors co-opting models originally developed for water and debris-flows with simplified physics, and combined them in a hueristic way allows different parts of an event to be modeled as one of two types of debris flows. The authors make a distinction between *viscous* debris flows, where clasts are supported by the viscosity of the flow, and *inertial* flows, where clasts readily settle out of the flow. Based on field evidence, the authors split the hydrograph of the event into five surges, which is each modeled individually, with individually calibrated parameters.

The authors assert that this technique captures the broad trends of erosion and deposition during this rainfall event, and further that this approach can be used to evaluate debris flow mitigation structures in similar arid environments. In particular, they highlight that the modeling approach allows one to test a range of different parameter values and debris flow types at a tolerable computational cost.

**Overall Comments**

Overall I found this manuscript to be reasonably clear, well-structured, and sound in its methods. I am not a frequent reader of NHESS, but from checking a few recently-published articles, this manuscript appears to be a good fit for the mission of the journal. The modeling approach the authors outline requires substantial tuning, user input, local knowledge, and has a limited ability to etrapolate beyond the location being evaluated. As a result, quite a bit of data would be required to use this modeling approach for hazard mapping or analyzing a proposed mitigation approach. That said, the authors have done a good job of describing and clearly outlining when their model is applicable and when it is not, which I think is an especially strong aspect of this paper. I came away with a good sense of where and when an approach like theirs would be applicable, and what data I would need to collect in advance to use their methods.

I see no major obstacles to publication, and have enumerated a few minor comments below that I think would improve the manuscript's clarity.

**Minor Comments**

1. Throughout, the authors use the term "telescopic-like" to describe the sediment deposits and landforms they are observing. While this term provides an approximate visual intuition, I think the authors could come up with something more precise to capture this.

2. Throughout, the authors italicize place names like *Crucecita Alta*, etc. If this is the journal's style, then so be it, but this is a strange convention that I don't care for.

3. Several times, the authors refer to "low-frequency, high-magnitude" debris flow events, but no reference values are given. Does low-frequency mean once a decade or once a century? It would be better to be specific here, and talk about the time-scales and magnitudes in dimensional terms that the reader can put in context (example at line 84).

4. The location for this field location is not clear from the manuscript as written. The authors give the location only as "~29°S, 70°W" which is very imprecise. I was eventually able to locate the fan in question by cross-referencing the figures in Cabré et al. 2020 (*Progress in Physical Geography: Earth and Environment*). The fan is located at: (28.895569°S, 70.449925°W). The authors should give this more precise location so that folks can locate this field site and look at aerial imagery in google Earth etc.

5. The authors discuss "sediment connectivity" at several points, but it is not *entirely* clear to me what they mean by this. Do they simply mean that sediment flows from the fan into the river? If so, it is important to talk about the time scale over which this is evaluated. In a mountainous catchment like this, the river will transport all of the sediment in the fan over geologic time (10s of thousands of years), so the fan is simply a temporary sediment storage place. However, if the authors are speaking only about short time-scales, then the connectivity is related to (a) how much sediment can bypass the fan and move directly into the river, plus (b) the amount of sediment the river can scavenge from the fan toe. I might recommend choosing a different term, because it sounds to me more like the authors are evaluating how much the sediment is "partitioned" between the topset of the fan versus how much spills into, and is carried away by the river.

**Figure comments**

1. FIGURE 1: I like this map a lot. It is well designed, but could the authors put latitude and longitude instead of UTM coordinates? This would help users locate the field site. Without knowing what UTM zone you are in, we cannot accurately locate this catchment. This idea goes for all of the maps.

2. FIGURE 2: These colors are not friendly to colorblind readers. I am not personally colorblind, but 5–8% of males are, and ~1% of females are. The potential for miscommunication could be easily avoided with a different color choice. I recommend the authors try a multi-panel figure here in addition to changing the color categories.

3. FIGURE 3: The caption uses "longitude" when I think the authors mean that the channel is 550 long.

4. FIGURE 5: It would be helpful to label the columns and rows of this set of maps since this is a matrix of conditions. Specifically, it would be helpful to have "Surge 1" and "Surge 2" on the left-hand side of the left panels, while "Max Flow Depth", and "Final Flow Depth" were above the top row. This will make it easier for your reader, who has to check back and forth from the caption at the moment.

5. FIGURE 6: The same panel labeling comment applies here and to all of your other multi-panel map figures. It would make it so much better for the reader.

6. FIGURE 7: Again, these color lines are not colorblind-friendly. Some readers will not be able to tell the green and red lines apart.

**Line comments**

1. "...in debris flow prone areas..." (no **s**)                                        line 35

2. missing reference?                                                                    line 50

3. "surrogated" is a strange word choice here. Consider "thus, it can be used as a surrogate."

4. "Riverbank erosion has previously trimmed the alluvial fan toe to the event" is an awk-        line 109
ward construction. Consider revising.

5. I do not see a supplement, but it would be good if this imagery were put into a supplementary information file for the reader to easily access. A screenshot would do   line 117

6. "it can surrogate the flow" → "it can **be a** surrogate **for** the flow"   line 173

7. Is this a typo? It seems you refer to $W_i^*$ being a function of the function before it is defined?   line 197

8. put this link to the data somewhere else, like a supplement or data availability statement. Does it have a DOI?   line 217

9. The authors say that low-frequency events are becoming more frequent. So are they still low-frequency? better to put an actual recurrence interval here.   line 461

---

## Author Comment (AC1)

**Reply on Referee Comment 1:**
**A modeling methodology to study the tributary-junction alluvial fan connectivity during a debris flow event**

November 2021

We thank the reviewer for his time in commenting on our paper. In this document, we answer each individual comment. For clarity, the reviewer's comments are in black, while our answers are in blue.

**Broad comments:**

Debris flow or debris flood? Please address the terminology issue also in the light of this new paper Church, Michael; Jakob, Matthias (2020). What is a debris flood?. Water Resources Research, doi:10.1029/2020WR027144

The debris flood description provided by Church and Jakob (2020) is similar to our preprint description for inertial debris flows. Church and Jakob (2020) recognize that debris floods rely on the tractive forces of the water. This is the key difference with viscous debris flows described in our preprint, where the tractive forces of the mud dominate (water and fine sediment mixture). Church and Jakob (2020) mention that "debris floods represent water-driven flood flows with high bedload transport of gravel to boulder size material and significant damage potential." They also highlight that the term "hyperconcentrated flows" may lead to confusion when talking about debris floods. Even though a particular flood may be both a debris flood and a hyperconcentrated flow, the generalization of the term "hyperconcentrated flows" overlooks an essential distinction between suspension and bedload dominance in the sediment transport process. Therefore we will eliminate the term "hyperconcentrated flows".

Surges 3 and 4 are described as inertial debris flows in our work. However, we recognize that inertial debris flows have a broad significance, where many types of flows are lumped. The term debris flood for surges 3 and 4 is, therefore, more specific to depict what our model is able to simulate. Furthermore, our work shows that collisions and turbulent stresses dominate in surges 3 and 4, which result in erosion and sedimentation that modify the fan's topography. The sediment transport characteristics of debris floods described in Church and Jakob (2020), therefore, support the assumption in our paper that surges 3 and 4 can be simulated as a Newtonian flow together

with a sediment transport model. On the other hand, viscous debris flows (surges 1 and 2) are properly defined as debris flows.

In conclusion, we will add the debris flood classification to the description of surges 3 and 4 and a short description of what a debris flood is.

L75: the Crucecita Alta deposit is described in bulk with other fans in the paper (Cabré 2020a). Figure 6 of that paper shows the inferred sediment concentrations during the event but, since we are focusing on this test site now, I would advise you to provide a paragraph discussing how you estimated the sediment concentrations for each surge.

In our model, the sediment concentration is a function $C_V(t)$, where $t$ is time, and it depends on the minimum and maximum plausible volumetric sediment concentrations. Zegers et al. (2020) present this function in detail. Specifically, in this article, we introduced a semi-automatic calibration algorithm (will be called Decision Support System in the future, see below). This algorithm evaluates the effect of different sediment concentrations and constrains the cases that reproduce the event flooded area and deposited volume. Therefore, sediment concentrations were not estimated directly but instead calibrated with our novel algorithm.

L75(continue): Moreover, in Cabré 2020a it is stated that the maximum thickness of the deposits is 100 cm while your simulations of the viscous debris flow show a larger thickness of deposits for Surge 1.

The resulting deposit thicknesses of the viscous debris flows (surges 1 and 2) are highly dependent on the Surface Detention parameter $(SD)$. In Line 325, we analyze the effect of the $SD$ parameter over the deposit thickness associated with Surge 2 but not for surge 1. We will add in this paragraph that Surge 1 resulted in similar behavior.

The problem with the $SD$ parameter is that the final deposited volume is very sensitive to this parameter. In Fig. 4. of Zegers et al. (2020), it is shown that SD corresponds to the parameter with the highest sensitivity for alluvial fans that buffer the sediment discharge. Also, it is complicated to select an $SD$ parameter value based on field data. We started with an $SD$ parameter value of 1 m resulting in slightly thicker deposits. Since we have no clarity on the correct value for $SD$, our approach was to test lower values of SD (0.5m and 0.7m). Still, unfortunately, the flow spreads over the surface, losing any correlation with the affected areas. Finally, we decided to keep the $SD$ parameter value equal to 1 m because this value could reproduce the event better than an SD value equal to 0.5 or 0.7 m. We tested a set of one hundred models for each $SD$ parameter value with our novel algorithm to find the rheological parameters.

L75(continue): Is the remoulding of the subsequent floods that reduces the deposit thickness in the upper part of the fan? Is it possible to add a map showing the deposit thickness inferred from the geomorphological survey?

It is possible that remoulding of the subsequent floods could have reduced the deposit thickness, but, this is very difficult to affirm based on our field observations. However, as stated in the previous answer, the overestimation of the deposit thickness can be a spurious effect that is analyzed in line 325 of the preprint.

In Figure 1 we show the point measurements of the deposit thicknesses taken on the field for

[Figure]

Figure 1: Point measurements of deposit thicknesses taken for F1 and F2 deposits.

F1 and F2 that we used to infer the mean deposit thickness.

The model calibration of debris flows just on the basis of the impacted area instead of deposits can lead to errors. In case it is not possible, please insert two sentences in the text highlighting this potential problem.

We based our methodology on the recent paper by Zegers et al. (2020). One of this study's main findings was that the calibration by flood area and deposited volume also constrained the flow velocity and flow height (Figure 5 in Zegers et al. (2020)). Therefore, in this study, we analyzed these two variables in a two-step procedure. First, we filtered by flood area indexes considering the correlation of the flooded pixels. Secondly, we analyzed the filtered cases by considering the resulting deposited volume and visual inspection, where we manually selected the best case out of the filtered runs.

Even though the flooded area filter returns a manageable amount of runs for visual inspection, we agree with the reviewer that including the volume filter directly to our Decision Support System (DSS) is a better approach. However, one must remember that the important goal of the DSS is to

find the best cases (for example, five out of one hundred) instead of finding specific thresholds for the volume and area filters. For instance, in order to obtain three runs for F1 (Fig. 3), a volume filter of $\pm 60\%$ was necessary. On the other hand, all cases in F2 filtered by area pass the volume filter of $\pm 60\%$ (Fig. 3). Therefore, it should be noted that this filter ($\pm 60\%$) is arbitrary, and it can change for every studied case.

We will add a short explanation of this to the document.

L259: I completely agree with you that for modelling calibration is sometimes difficult or not robust to rely only on a automatic algorithms to select the best fit parameters; it is in fact usually best to incorporate the expert knowledge euristic to select the best fit parameters and call the algorithm a Decision Support System rather than a automatic calibration algorithm. To better explain this to the readers you can provide a figure with the cloud points of your optimization indexes to show how clustered or nonclustered your 100 simulations were, so as to also understand how arbitrary is your "five runs" threshold.

We thank the reviewer for this helpful suggestion. We will change the algorithm's name to Decision Support System (DSS) since this name is closer to the actual capabilities of the algorithm. As the article and the reviewer highlight, this algorithm constrains the search for the rheological parameters that reproduce the event. However, the user has the final decision about which model is the best.

Thanks to this suggestion, we found an undesirable behavior in our Decision Support System (DSS). Even though the parameter sampling method distributed the rheological parameters within plausible value ranges, their combination could result in unrealistic values for $\tau$ and $\eta$. As a result, only 56 out of 100 cases have reasonable yield stress and dynamic viscosity values. We corrected this problem by adding a step where we check if $\tau$ and $\eta$ have reasonable values. If they don't, new values of $\beta_1$ and $\beta_2$ and $C_V^{max}$ are set until we obtain values of $\tau$ and $\eta$ lower than $50000\ dyn\ cm^{-2}$ and $100000\ Pa$, respectively.

To be consistent with our population of 100 cases, we re-run our simulations with the corrected algorithm. In Figure 2, we present the distribution of the 100 runs for the area "True Positive" ($Area^{TP}$) vs. the area "False Positive" ($Area^{FP}$). These areas were defined in the preprint. In Fig 2.a, we present the old set of 100 runs where black stars correspond to values of $\tau$ and $\eta$ higher than the maximum plausible values. Red stars correspond to the 56 cases with reasonable values for $\tau$ and $\eta$. Figure 2.b presents a new set of 100 runs where the algorithm was corrected as explained previously. From both distributions, we observe that the red cases in Fig 2.a have the same distribution as the cases in Fig 2.b. The only difference now is that instead of having 56 runs with plausible values, we have 100. Consequently, the runs that now pass the flooded area filters are ten instead of four for F1 and nine instead of four for F2.

In Figures 3.a and 3.b, we show maps of the areas considered as "Observed positive" ($Area_{OP}$) and "Observed negative" ($Area_{ON}$) for the facies F1 and F2, respectively. In addition, in Figures 3.c and 3.d, we present the distribution of $Area^{TP}$ vs. $Area^{FP}$ for surges 1 and 2 associated with F1 and F2, respectively. Our approach with the DSS is to find a manageable amount of runs for visual inspection that better reproduce the area and volume. Therefore, we looked for the runs that maximize $Area^{TP}$ and minimize $Area^{FP}$. For F1 (Fig. 3.c), for example, the best ten cases have an $Area^{TP} > 70\%\ Area_{OP}$ and an $Area^{FP} < 30\%\ Area_{OP}$. For F2 (Fig. 3.d), the best nine

[Figure]

Figure 2: (a) the original set with $\tau$ and $\eta$ values higher than the maximum plausible value. (b) the new set of runs with values of $\tau$ and $\eta$ corrected to remain within a reasonable range of values.

cases have an $Area^{TP} > 90\%$ $Area_{OP}$ and an $Area^{FP} < 10\%$ $Area_{OP}$. In Fig. 3, the green hatched areas show the runs that pass the area filtering. It is important to note that the area thresholds for surge 1 and surge 2 are different due to their dissimilar distribution of $Area^{TP}$ vs. $Area^{FP}$. On the other hand, the cases that pass the volume filter are presented with blue stars (Fig. 3). Finally, **filtered runs by area and volume correspond to blue stars within the green hatched area.**

For F1, $Area^{TP}$ ranges from 0.3 to almost 1.0, whereas $Area^{FP}$ ranges from 0.05 to 1.4. For F2, $Area^{TP}$ ranges from 0.85 to almost 1.0, whereas $Area^{FP}$ ranges from 0.05 to 0.25.

About the data distribution, we found that, for F1, the points are spread, whereas for F2 are mainly clustered. The broader range of $Area^{TP}$ in F1 than in F2 is due to the higher $C_V^{mean}$. For F1, the inflow volume of water is less than for surge 2, but the volume of sediment is larger than for surge 2. As a result, F1 has higher $C_V^{mean}$ giving the ability for this surge to stop before filling $Area_{OP}$. On the other hand, the more diluted nature of surge 2 makes it easier for this surge to flood $Area_{OP}$.

In the maps of Figure 3, we can see a crucial difference between F1 and F2 cases. In F1, the debris flows do not reach the river, so the area downstream of F1 is considered "observed negative". Contrary, since F2 is connected with the river, $Area_{ON}$ was selected only at both sides of the mapped deposit. This difference results in a broader range of $Area^{FP}$ for surge 1 than $Area^{FP}$ for surge 2.

[Figure]

Figure 3: Correct vs. incorrect reproduced areas by one hundred runs tested by our Decision Support System.

Since the set of 100 runs is new, we added the original case used in our model to Figure 3.c and 3.d as a gray square. As seen here, the old selectedruns meet both the area and the volume filter. Therefore, it is not necessary to rerun the calibrated model. Moreover, the distribution of the results in Fig. 2 did not change due to the additional step in the DSS algorithm.

**Specific Comments:**

L10: if a channel is wide enough
We thank the reviewer's suggestions, we will replace it with: "if a channel is large enough".

L50: citation missing (?)
We apologize for the citation misspelling. It corresponds to Takahashi 2014 (book).

Figure 1: it is not clear what do the black lines represent in part b. Can you show in a less stylized way where exactly the element at risk (roads and buildings) are located in fig b?
We present a less stylized way in Figure 4.

L386: to design mitigation works
We thank the reviewer's suggestions, the word "test" will be replaced with "design".

L389: the broad flow typology
We thank the reviewer's suggestions, we will rephrase the line to "The same hydraulic works should be studied under the broad flow typology".

L398: are deviated and forced to deposit
We thank the reviewer's suggestions, we will rephrase the line to "are deviated and forced  to deposit due to the previous deposits".

L399: With surge 4 avulsion is present, inundating the southern portion of the fan
We thank the reviewer's suggestions, we will rephrase the line to " With surge 4, avulsion is present, inundating the southern portion of the fan".

Figure 8: please explain why the deposition pattern in surge 4 is so rectified, it seems a little bit unrealistic - did you experience problems with DTM interpolation? Can you show the contour lines of your model DTM topography?
In Figure 5, we present Figure 3 of the preprint with the contour lines every 2 m added. We did not experience any problems with DTM interpolation, but it should be noted that the model has a 5 $m/pixel$ resolution. FLO-2D has a grid of square cells, but these cells are also connected diagonally (Figure 6). Thus, the flow has eight possible flow direction. When the flow aligns with one of these directions, the solver tends to generate preferential flows following this alignment. As seen in Figure 8 in the preprint, these rectified patterns in surge 4 align in near 45°(south-west direction).

[Figure]

Figure 4: New figure with Google satellite imagery.

[Figure]

Figure 5: Contour lines every 2m of the topography.

[Figure]

Figure 6: FLO-2D square cells and their eight flow connections between cells.

L435: buffer?

Fryirs et al. (2007) named the lateral impediment of sediment conveyance as a buffer, and the longitudinal impediment of sediment conveyance was named as a barrier. Since surge 1 could not reach the river, it is possible to say that the alluvial fan completely buffers this viscous debris flow.

*Fryirs, K. A., Brierley, G. J., Preston, N. J., & Spencer, J. (2007). Catchment-scale (dis)connectivity in sediment flux in the upper Hunter catchment, New South Wales, Australia. Geomorphology, 84(3-4), 297-316.*

L441-446: please revise this paragraph as the concepts expressed are clear but their formulation is not so

Original paragraph:

"We also studied how the Crucecita Alta alluvial fan would react to a channel. This approach has proven helpful because the channel (mitigation work) was expected to reduce affected areas on the alluvial fan and enhance the trunk river obstruction. However, the results in scenario 1 indicate that a channel reduces affected areas and avoids headward erosion. Thus, it reduces the amount of sediment that finally reaches the river reducing the river obstruction. In scenario 2, even though the affected areas are reduced, deposition occurs between the apex and the fan's mid zone and, consequently, a channel overflow. However, headward erosion is also avoided for scenario 2, and the new lobe at the fan toe reduces its size considerably."

Proposed new paragraph:

Our methodology is helpful to study hypothetical scenarios, such as a channel designed to mitigate affected areas but increasing the coupling status of the alluvial fan. The results of scenario 1 (Fig. 8.a to 8.d) show that viscous debris flows can reach the river since they remain confined to the channel. However, sediment deposition occurring during the flow reduces the channel transport capacity for following surges. Thus, when the inertial debris flow surges occur, we observe that the channel can convey surge 3 but not surge 4. Compared to the original 25M event, smaller erosion areas result in lower sediment loads, reducing the size of the new lobe at the fan's toe. The results of scenario 2 (Fig. 8.e and 8.f), which consider only the inertial debris flows, show that the channel avoids the headward erosion observed in the 25M event. Consequently, the amount of sediment deposited at the new lobe at the fan toe is reduced considerably compared to the original case. However, we expected that the presence of the channel would increase the structural connectivity.

L484: incide

We will change the word "incise" to "carve" in order to avoid confusion.

---

## Author Comment (AC2)

**Reply on Referee Comment 2:**
**A modeling methodology to study the tributary-junction alluvial fan connectivity during a debris flow event**

**November 2021**

We thank the reviewer for his time in commenting on our paper. In this document, we answer his comments. For clarity, the reviewer's comments are in black, while our answers are in blue.

**1 Minor comments**

1. Throughout, the authors use the term "telescopic-like" to describe the sediment deposits and landforms they are observing. While this term provides an approximate visual intuition, I think the authors could come up with something more precise to capture this.

The term "telescopic-like" deposit has been used for a long time (Blissenbach, 1954). Colombo (2005) provided a thorough analysis of tributary-junction alluvial fan's geometry in the Argentine Andean Ranges. The alluvial fans described by this author have similar geometry to Crucecita Alta. Colombo (2005) concludes that these telescopic-like deposit morphologies (terraced morphologies) are not associated with regional base levels, tectonic activity, or climatic changes. Instead, the genesis of the telescopic-like morphologies can be explained by the El Niño Southern Oscillation (ENSO). Furthermore, Cabré et al. (2020) characterized the role of telescopic-like deposits in the coupling mechanisms between alluvial fans and the trunk valley. Changes in the surges rheology, for example, produce a shift in the alluvial fan's coupling status from buffers to couplers that allow the transmission of sediment down-system.

We decided to keep the term "telescopic-like" due to their geometric and sedimentological significance (Colombo, 2005; Cabré et al., 2020).

2. Throughout, the authors italicize place names like Crucecita Alta, etc. If this is the journal's style, then so be it, but this is a strange convention that I don't care for.

We thank the reviewer's comment. According to the NHESS house standards, foreign words are italicized, but this does not apply to proper nouns. We will correct this in the manuscript.

3. Several times, the authors refer to "low-frequency, high-magnitude" debris flow events, but no reference values are given. Does low-frequency mean once a decade or once a century? It would be better to be specific here, and talk about the time-scales and magnitudes in dimensional terms that the reader can put in context (example at line 84).

We thank the reviewer's comment. We will add the following analysis in the Study Area section:

It is complicated to assert a specific recurrence interval for these events due to the lack of data. However, there has been a consensus that storms like the 25M event in the Atacama Desert occur once a century (Ortega et al., 2019). Thus, we considered a "low-frequency" event as an event that occurs, on average, once a century. About the magnitude of the event, Aguilar et al. (2020) estimated the mean erosion rate of the Huasco basin during the 25M event equal to 1.3 mm for the entire catchment. On the other hand, Aguilar et al. (2014) estimated erosion rates of $0.03 - 0.08\ mm\ yr^{-1}$ during the last $6 - 10\ Myr$ for the same catchment. Considering that an event such as the 25M event occurs once a century, we can say that the 25M event has a "high-magnitude" because this single event contributed 15% up to 40% of the total eroded volume every one hundred years.

4. The location for this field location is not clear from the manuscript as written. The authors give the location only as "$\sim 29°S, 70°W$" which is very imprecise. I was eventually able to locate the fan in question by cross-referencing the figures in Cabré et al. 2020 (Progress in Physical Geography: Earth and Environment). The fan is located at: $(28.895569°S, 70.449925°W)$. The authors should give this more precise location so that folks can locate this field site and look at aerial imagery in Google Earth, etc.

We have added the specific location suggested by the reviewer to the Figure 1 caption. Also, we changed all our maps from UTM coordinates to lat-log coordinates so anyone can find the place of the alluvial fan without knowing the UTM zone.

5. The authors discuss "sediment connectivity" at several points, but it is not *entirely* clear to me what they mean by this. Do they simply mean that sediment flows from the fan into the river? If so, it is important to talk about the time scale over which this is evaluated. In a mountainous catchment like this, the river will transport all of the sediment in the fan over geologic time (10s of thousands of years), so the fan is simply a temporary sediment storage place. However, if the authors are speaking only about short time-scales, then the connectivity is related to (a) how much sediment can bypass the fan and move directly into the river, plus (b) the amount of sediment the river can scavenge from the fan toe. I might recommend choosing a different term, because it sounds to me more like the authors are evaluating how much the sediment is "partitioned" between the topset of the fan versus how much spills into, and is carried away by the river.

This study captures how a fan-river system evolves during a single storm event. The time scale in which this study focuses corresponds to the length of the 25M event, i.e., $\sim$ 48 - 72 $h$. About the spatial scale, it corresponds to a specific tributary-junction alluvial fan. Tributary-junction alluvial fans have a spatial scale around $10^1 - 10^3\ m$ (Mather et al., 2017). The development of the "sediment connectivity" characteristics in this time scale ($\sim$ days) has been studied in multiple studies (Aguilar et al., 2020; Cabré et al., 2020). Aguilar et al. (2020) characterize the connectivity of the whole catchment as the efficiency of the system to transport sediment down-system during the 25M event. Cabré et al. (2020) proposed a conceptual model to explain the changes in connectivity due to extreme storm events. Heckmann et al. (2018) reviewed the main indices of "sediment connectivity" which apply to a specific location within the fluvial system. Therefore, the concept of "sediment connectivity" or "(dis)connectivity" implies a balance between top-down and bottom-up processes (see Mather-etal2017 for a comprehensive review). Top-down processes are controlled by climatic and geological variability that directly impact the sediment

supply. On the other hand, bottom-up processes are controlled by morphological characteristics such as base-level driven processes. Finally, a balance between these two processes determines the fan-river system's "connectivity" or coupling status.

That said, the reviewer is right when he says that all of the sediment will be transported over geologic time. In fact, Cabré et al. (2017) studied the sedimentary infilling of the Huasco catchment using $^{14}C$ samples. They demonstrated that alluvial fans and terraces in the valley are younger than 15000 years, and they conclude that the infilling started at 11000 years BP, finishing about 2000 years BP.

**2 Figure comments**

Figure 1: I like this map a lot. It is well designed, but could the authors put latitude and longitude instead of UTM coordinates? This would help users locate the field site. Without knowing what UTM zone you are in, we cannot accurately locate this catchment. This idea goes for all of the maps.

We have changed all our maps to EPSG 4326, WGS84. The changes in the figures presented here also consider the changes suggested by RC1.

[Figure]

Figure 1: Study Area. (a) *Crucecita Alta* catchment $(28.895569°S, 70.449925°W)$. The grey polygon is the river segment that includes the studied fan topography section used in the numerical model. The unfilled polygon depicts the *Crucecita Alta* catchment ($13\ km^2$). (b) *Crucecita Alta* alluvial fan main geometric features where the feeder-channel generated during the 25M event are presented with dotted lines.

Figure 2: These colors are not friendly to colorblind readers. I am not personally colorblind, but 5-8% of males are, and ∼ 1% of females are. The potential for miscommunication could be easily avoided with a different color choice. I recommend the authors try a multi-panel figure here in addition to changing the color categories.

We have changed the color palette of all our figures to this color-blind-friendly color palette retrieved from `https://www.color-hex.com/color-palette/49436`. We also accepted the suggestion of a multi-panel figure.

[Figure]

Figure 2: Available data for 25M event in *Crucecita Alta* alluvial fan. (a) The facies F1, F2, F3, F4, and F5 were retrieved from Cabré et al. (2020) and LiDAR topography surveyed by IDIEM (2019). (b) Flow hydrograph obtained from the hydrologic model performed by IDIEM (2019). The colors used to identify the facies in (a) depict their correlation with the surges in (b).

Figure 3: The caption uses "longitude" when I think the authors mean that the channel is 550 long.

[Figure]

Figure 3: Topographic data modifications. Post-event topography corresponds to the available LiDAR topography, while the pre-event topography is a restitution based on satellite images and the available topography. The synthetic channel (brown dashed line) attached to the LiDAR topography is 550 $m$ long while feeder channel (orange dashed line) is 450 $m$ long. In the post-event topography, the feeder channel results from the inertial debris flow incisions on the alluvial fan. In the mitigation works topography, a straight rectangular channel replaces the feeder channel.

Figure 5: It would be helpful to label the columns and rows of this set of maps since this is a matrix of conditions. Specifically, it would be helpful to have "Surge 1" and "Surge 2" on the left-hand side of the left panels, while "Max Flow Depth", and "Final Flow Depth" were above the top row. This will make it easier for your reader, who has to check back and forth from the caption at the moment.

We have implemented the changes suggested by the reviewer. Also, we changed the color map of our results from "Spectral_r" to "RdBu_r" to avoid having contrasts between red/orange colors and the green color.

[Figure]

Figure 4: Viscous debris flow surges. Top panels correspond to the results for surge 1, whereas bottom panels correspond to surge 2. Dashed polygons show the extent of mapped facies F1 and F2. Left panels present the maximum flow depth, whereas right panels present the final flow depth at the end of each surge. Topographic changes for subsequent surges are updated based on the final flow depth of the previous surge.

Figure 6: The same panel labeling comment applies here and to all of your other multi-panel map figures. It would make it so much better for the reader.

[Figure]

Figure 5: Inertial debris flow surges. Top panels correspond to the results for surge 3, whereas bottom panels correspond to surge 4. Dashed polygons delimit the telescopic-like deposit, i.e., the incision in the alluvial fan and the new lobe at the fan toe. Left panels present the maximum flow depth, while right panels present the topographic change for erosion (negative values) and deposition (positive values).

Figure 7: Again, these color lines are not colorblind-friendly. Some readers will not be able to tell the green and red lines apart.

We also implemented the new colorblind-friendly color palette in this figure, maintaining the relationship between facies and colors consistent.

[Figure]

Figure 6: Morphological evolution of the tributary-junction alluvial fan. (a) Location of the longitudinal sections presented in b and c. (b) Longitudinal profile A-A' of the alluvial fan topography after each surge. (c) Longitudinal profiles B-B', C-C', D-D' of the river's topographic evolution.

Figure 8

We implemented all previous changes in Figure 8 of the preprint

[Figure]

Figure 7: Two future scenarios are modelled to understand the possible effects of a channel (mitigation works): First scenario (a,b,c,d) and second scenario (e,f).

**3   Line comments**

1. "...in debris flow prone areas..." (no s) line 35
We thank the reviewer's suggestion. We will change the line to: "For example, in debris flows prone areas,.."

2. missing reference? line 50
We apologize for the citation misspelling. It corresponds to Takahashi (2014)

3. "surrogated" is a strange word choice here. Consider "thus, it can be used as a surrogate."
We thank the reviewer's suggestion. We will replace "surrogated" as the reviewer suggests.

4. "Riverbank erosion has previously trimmed the alluvial fan toe to the event" is an awkward construction. Consider revising. line 109
We thank the reviewer's suggestion. We will replace the sentence with "Previous to the event, riverbank erosion trimmed the alluvial fan toe."

5. I do not see a supplement, but it would be good if this imagery were put into a supplementary information file for the reader to easily access. A screenshot would do line 117
We thank the reviewer's suggestion. We will add a supplement file with screenshots of google satellite imagery of Crucecita Alta fan for 2013, 2016, and 2017.

6. "it can surrogate the flow" -¿ "it can be a surrogate for the flow" line 173
We thank the reviewer's suggestions. We will change this line to "it can be a surrogate for the flow rheology".

7. Is this a typo? It seems you refer to $W_i^*$ being a function of the function before it is defined? line 197
Parker et al. (1982) proposed the dimensionless sediment transport rate $W^*$ as a function of $\phi_{50}$ (equation (7) of the preprint). Parker & Klingeman (1982) modified this equation to subdivide the transport rate into fractions. Thus, the dimensionless fractional transport rate $W_i^*$ is associated with a specific sediment size diameter $i$. To this purpose, $W_i^*$ is also calculated with equation (7) but using $\phi_i$ instead of $\phi_{50}$. Equation (9) of the preprint defined $\phi_i$.

8. put this link to the data somewhere else, like a supplement or data availability statement. Does it have a DOI? line 217
We will add the topographic data to a supplement.

9. The authors say that low-frequency events are becoming more frequent. So are they still low-frequency? better to put an actual recurrence interval here. line 461
Estimating the recurrence interval of these events is one of the critical problems that hazard assessment projects in Chile are facing. Debris flow hazard matrices such as the one proposed by Hürlimann et al. (2008) require an estimation of the intensity and the probability of occurrence of debris flows. However, it is very uncertain to estimate the probability of occurrence. The triggering of debris flows in this arid system is related to climatic, hydrologic, and geological characteristics, but no standard procedure exists to estimate the recurrence interval.

In this arid system, Aguilar et al. (2020) showed that catchments are transport-limited, i.e., the more water flows, the more sediment is transported. On the other hand, Ortega et al. (2019) showed that the recurrence of extreme rainfall events has increased in the last century, but the mean annual volume of rain has decreased. This condition is unfortunate because the lack of mild-to-normal rains (i.e., rains with two years of return periods) enhance sediment storage. Consequently, when an extreme storm event occurs (i.e. rains with one hundred return period), there is a more significant amount of sediment available. Since extreme rainfall events are the main responsible for debris flow triggering in the Atacama Desert, one may guess that debris flow events are becoming more frequent.

**References**

Aguilar, G., Cabré, A., Fredes, V., & Villela, B. (2020). Erosion after an extreme storm event in an arid fluvial system of the southern Atacama Desert: an assessment of the magnitude, return time, and conditioning factors of erosion and debris flow generation. *Natural Hazards and Earth System Sciences*, 20(5), 1247–1265.

Aguilar, G., Carretier, S., Regard, V., Vassallo, R., Riquelme, R., & Martinod, J. (2014). Grain size-dependent 10Be concentrations in alluvial stream sediment of the Huasco Valley, a semi-arid Andes region. *Quaternary Geochronology*, 19, 163–172.

Blissenbach, E. (1954). Geology of alluvial fans in semiarid regions. *Geological Society of America Bulletin*, 65(2), 175–190.

Cabré, A., Aguilar, G., Mather, A. E., Fredes, V., & Riquelme, R. (2020). Tributary-junction alluvial fan response to an ENSO rainfall event in the El Huasco watershed, northern Chile. *Progress in Physical Geography*.

Cabré, A., Aguilar, G., & Riquelme, R. (2017). Holocene evolution and geochronology of a semiarid fluvial system in the western slope of the central andes: Ams 14c data in el tránsito river valley, northern chile. *Quaternary International*, 438, 20–32.

Colombo, F. (2005). Quaternary telescopic-like alluvial fans, Andean Ranges, Argentina. *Geological Society, London, Special Publications*, 251(1), 69–84.

Heckmann, T., Cavalli, M., Cerdan, O., Foerster, S., Javaux, M., Lode, E., Smetanová, A., Vericat, D., & Brardinoni, F. (2018). Indices of sediment connectivity: opportunities, challenges and limitations. *Earth-Science Reviews*, 187, 77–108.

Hürlimann, M., Rickenmann, D., Medina, V., & Bateman, A. (2008). Evaluation of approaches to calculate debris-flow parameters for hazard assessment. *Engineering Geology*, 102(3-4), 152–163.

IDIEM (2019). *Diseño de obras fluviales y de control aluvional, cuenca del río El Carmen. Informe Final. Volumen IV:Estudios Hidráulicos.* Technical Report 1.189.262, Investigación, Desarrollo e Innovación de Estructuras y Materiales - (IDIEM).

Mather, A., Stokes, M., & Whitfield, E. (2017). River terraces and alluvial fans: The case for an integrated Quaternary fluvial archive. *Quaternary Science Reviews*, 166, 74–90.

Ortega, C., Vargas, G., Rojas, M., Rutllant, J. A., Muñoz, P., Lange, C. B., Pantoja, S., Dezileau, L., & Ortlieb, L. (2019). Extreme ENSO-driven torrential rainfalls at the southern edge of the Atacama Desert during the Late Holocene and their projection into the 21th century. *Global and Planetary Change*, 175(February), 226–237.

Parker, G. & Klingeman, P. C. (1982). On why gravel bed streams are paved. *Water Resources Research*, 18(5), 1409–1423.

Parker, G., Klingeman, P. C., & McLean, D. G. (1982). Bedload and size distribution in paved gravel-bed streams. *Journal of the Hydraulics Division-Asce*, 108(4), 544–571.

Takahashi, T. (2014). *Debris flow: mechanics, prediction and countermeasures.* CRC Press/Balkema - Taylor & Francis Group, 2nd edition.

---

## Author Response (AR1)

**Author's Response:**
**A modeling methodology to study the tributary-junction alluvial fan connectivity during a debris flow event**

**December 2021**

The authors thank reviewers' valuable comments. Changes have been made to the manuscript to include reviewers' suggestions ("CrucecitaMS_V2.pdf"). To easily identify the changes done, we also uploaded a manuscript with tracked changes ("CrucecitaMS_V2_marked.pdf"). When necessary, we provide here line numbers where we included the reviewers' suggestions. First, we indicate the line number of the non-marked manuscript and, secondly, the line number of the manuscript with tracked changes. Also, the authors' responses in this document are marked in blue.

**Referee 1:**

Debris flow or debris flood? Please address the terminology issue also in the light of this new paper Church, Michael; Jakob, Matthias (2020). What is a debris flood?. Water Resources Research, doi:10.1029/2020WR027144

We thank the reviewer's observation. We have included the debris flood classification to surges 3 and 4 (section 2.2; lines 143-146; lines 160-163 marked version).

The Crucecita Alta deposit is described in bulk with other fans in the paper (Cabré 2020a). Figure 6 of that paper shows the inferred sediment concentrations during the event but, since we are focusing on this test site now, I would advise you to provide a paragraph discussing how you estimated the sediment concentrations for each surge.

In our model, the sediment concentration is a function $C_V(t)$, where $t$ is time, and it depends on the minimum and maximum plausible volumetric sediment concentrations. This article introduced a semi-automatic calibration algorithm (Decision Support System). This algorithm evaluates the effect of different sediment concentrations and constrains the cases that reproduce the event flooded area and deposited volume. Therefore, sediment concentrations were not estimated directly but instead calibrated with our novel algorithm. The resulting concentrations are presented in Table 1 together with the other calibrated parameters.

Moreover, in Cabré 2020a it is stated that the maximum thickness of the deposits is 100 cm while your simulations of the viscous debris flow show a larger thickness of deposits for Surge 1.

We analyze the effect of the $SD$ parameter over the deposit thicknesses for the viscous debris flow surges and the problem noticed by the reviewer in section 4.1; lines 338-345; lines 363-372 marked version.

The model calibration of debris flows just on the basis of the impacted area instead of deposits can lead to errors. In case it is not possible, please insert two sentences in the text highlighting this potential problem.

We thank the reviewer's comment. We have explained more specifically how our DSS considers the affected area and deposited volume to find the appropriate rheological parameters (section 3.4; lines 294-305; lines 316-328 marked version).

if a channel is wide enough
We thank the reviewer's suggestions, we replaced it with: "if a channel is large enough" (line 10; line 11 marked version).

citation missing (?)
We apologize for the citation misspelling. It corresponds to Takahashi 2014 (book) (section 1; line 46; line 54 marked version).

Figure 1: it is not clear what do the black lines represent in part b. Can you show in a less stylized way where exactly the element at risk (roads and buildings) are located in fig b?

We thank the reviewer's suggestions. We modified Figure 1 part b as suggested.

to design mitigation works
We thank the reviewer's suggestions, the word "test" was replaced with "design" (section 4.3; line 402; line 429 marked version)

L389: the broad flow typology
We thank the reviewer's suggestions, we rephrased the line to "The same hydraulic works should be studied under the broad flow typology" (section 4.3; line 404; line 431).

L398: are deviated and forced to deposit
We thank the reviewer's suggestion, we rephrased the line to "are deviated and forced to deposit due to the previous deposits" (section 4.4; line 413; lines 440-441).

With surge 4 avulsion is present, inundating the southern portion of the fan
We thank the reviewer's suggestions, we rephrased the line to "With surge 4, avulsion is present, inundating the southern portion of the fan" (section 4.4; lines 413-414; lines 441-442).

Figure 8: please explain why the deposition pattern in surge 4 is so rectified, it seems a little bit unrealistic - did you experience problems with DTM interpolation? Can you show the contour lines of your model DTM topography?

We have added contour lines every 2 m in Figure 3. We did not experience any problems with DTM interpolation, but it should be noted that the model has a $5\ m\ per\ pixel$ resolution. FLO-2D has a grid of square cells, but these cells are also connected diagonally. Thus, the flow has eight

possible flow directions. When the flow aligns with one of these directions, the solver tends to generate preferential flows following this alignment. As seen in Figure 8, these rectified patterns in surge 4 align in near 45°(south-west direction).

buffer?
Fryirs et al. (2007) named the lateral impediment of sediment conveyance as a buffer, and the longitudinal impediment of sediment conveyance was named as a barrier. Since surge 1 could not reach the river, it is possible to say that the alluvial fan completely buffers this viscous debris flow.

please revise this paragraph as the concepts expressed are clear but their formulation is not so
We thank the reviewer's suggestions. We rewrote the paragraph (section 5.1; lines 456-464; lines 485-499 marked version).

incide
We changed the word "incise" for "carve" in order to avoid confusion (section 6; line 501; line 542 marked version).

**Referee 2:**

1. Throughout, the authors use the term "telescopic-like" to describe the sediment deposits and landforms they are observing. While this term provides an approximate visual intuition, I think the authors could come up with something more precise to capture this.
    We decided to keep the term "telescopic-like" due to their geometric and sedimentological significance (Colombo, 2005; Cabré et al., 2020).

2. Throughout, the authors italicize place names like Crucecita Alta, etc. If this is the journal's style, then so be it, but this is a strange convention that I don't care for.
    We thank the reviewer's comment. We corrected this issue throughout the manuscript.

3. Several times, the authors refer to "low-frequency, high-magnitude" debris flow events, but no reference values are given. Does low-frequency mean once a decade or once a century? It would be better to be specific here, and talk about the time-scales and magnitudes in dimensional terms that the reader can put in context (example at line 84).
    We thank the reviewer's comment. We added a paragraph about time-scales and magnitudes associated with the 25M event (section 2.2; lines 126-131; lines 143-148 marked version).

4. The location for this field location is not clear from the manuscript as written. The authors give the location only as "$\sim 29°S, 70°W$" which is very imprecise. I was eventually able to locate the fan in question by cross-referencing the figures in Cabré et al. 2020 (Progress in Physical Geography: Earth and Environment). The fan is located at: $(28.895569°S, 70.449925°W)$. The authors should give this more precise location so that folks can locate this field site and look at aerial imagery in Google Earth, etc.
    We have added the specific location suggested by the reviewer to the Figure 1 caption. Also, we changed all our maps from UTM coordinates to lat-log coordinates so anyone can find the place

of the alluvial fan without knowing the UTM zone.

Figure Comments
We thank the reviewer's suggestions. These comments improved all figures in the manuscript.

1. "...in debris flow prone areas..." (no s)
We thank the reviewer's suggestion. We eliminated the s (section 1; line 38; line 39 marked version).

2. missing reference?
We apologize for the citation misspelling. It corresponds to Takahashi 2014 (book) (section 1; line 46; line 54 marked version).

3. "surrogated" is a strange word choice here. Consider "thus, it can be used as a surrogate."
We thank the reviewer's suggestion. We replaced it as suggested (section 1; line 54-55; line 62-63 marked version).

4. "Riverbank erosion has previously trimmed the alluvial fan toe to the event" is an awkward construction. Consider revising.
We thank the reviewer's suggestion. We rephrased the sentence (section 2; line 112; lines 128-129 marked version).

5. I do not see a supplement, but it would be good if this imagery were put into a supplementary information file for the reader to easily access. A screenshot would do.
We thank the reviewer's suggestion. We added a supplement file with screenshots of Google Earth satellite imagery of Crucecita Alta fan for 2013, 2016, and 2017. `https://doi.org/10.6084/m9.figshare.c.5739287.v1`

6. "it can surrogate the flow" -¿ "it can be a surrogate for the flow"
We thank the reviewer's suggestions. We changed this line to "it can be a surrogate for the flow rheology" (section 3.1; lines 183-184; lines 200-201 marked version).

8. put this link to the data somewhere else, like a supplement or data availability statement. Does it have a DOI? line 217
We added the topographic data to a supplement. the doi is `https://doi.org/10.6084/m9.figshare.c.5739287.v1`

**Final remarks**

We added a paragraph in the introduction to present the research question this work addresses explicitly. Additionally, we moved the previous paragraph to introduce the research question appropriately (section 1; lines 67-76; 78-87 marked version).

We reduced the conclusion section's length to communicate this study's main findings more effectively (section 6).